# Differential human antibody repertoires following Zika infection and the implications for serodiagnostics and disease outcome

Supriya Ravichandran[1], Megan Hahn[1], Pablo F. Belaunzarán-Zamudio[2], José Ramos-Castañeda[3], Gabriel Nájera-Cancino[4], Sandra Caballero-Sosa[5], Karla R. Navarro-Fuentes[6], Guillermo Ruiz-Palacios[7], Hana Golding[1], John H. Beigel [8,9] & Surender Khurana[1]

Zika virus (ZIKV) outbreak in Americas led to extensive efforts to develop vaccines and ZIKV-specific diagnostics. In the current study, we use whole genome phage display library spanning the entire ZIKV genome (ZIKV-GFPDL) for in-depth immune profiling of IgG and IgM antibody repertoires in serum and urine longitudinal samples from individuals acutely infected with ZIKV. We observe a very diverse IgM immune repertoire encompassing the entire ZIKV polyprotein on day 0 in both serum and urine. ZIKV-specific IgG antibodies increase 10-fold between day 0 and day 7 in serum, but not in urine; these are highly focused on prM/E, NS1 and NS2B. Differential antibody affinity maturation is observed against ZIKV structural E protein compared with nonstructural protein NS1. Serum antibody affinity to ZIKV-E protein inversely correlates with ZIKV disease symptoms. Our study provides insight into unlinked evolution of immune response to ZIKV infection and identified unique targets for ZIKV serodiagnostics.

[1] Division of Viral Products, Center for Biologics Evaluation and Research (CBER), FDA, Silver Spring, MD 20993, USA. [2] Departamento de Infectología, Instituto Nacional de Ciencias Médicas y Nutrición Salvador Zubirán, Mexico City 14080, Mexico. [3] Instituto Nacional de Salud Publica, Cuernavaca 62100, Mexico. [4] Hospital Regional de Alta Especialidad Ciudad Salud, Tapachula 30830 Chiapas, Mexico. [5] Instituto de Seguridad y Servicios Sociales de los Trabajadores del Estado, Tapachula 30740 Chiapas, Mexico. [6] Instituto Mexicano del Seguro Social, Tapachula 30700 Chiapas, Mexico. [7] Comisión Coordinadora de los Institutos Nacionales de Salud y Hospitales de Alta Especialidad, Ministry of Health, Mexico City 14080, Mexico. [8] Leidos Biomedical Research, Inc., Frederick National Laboratory for Cancer Research, Frederick, MD 21701, USA. [9] Present address: National Institute of Allergy and Infectious Diseases, National Institutes of Health, Bethesda, MD 20852, USA. Correspondence and requests for materials should be addressed to S.K. (email: Surender.Khurana@fda.hhs.govkkk)

Since its discovery in 1947[1,2], Zika virus (ZIKV) was associated primarily with sporadic infections in humans with mild symptoms. However, during the recent 2015–2016 outbreak in Latin America, ZIKV infections were associated with developmental and neurological complications including microcephaly in newborn babies and Guillain-Barré syndrome in adults[3–7]. This has prompted a major emphasis on vaccine development[8–13], and ZIKV-specific monoclonal antibodies (MAbs) or drugs with therapeutic/preventive potential and low risk of antibody-dependent enhancement (ADE)[14–19]. In addition, accurate diagnostics of ZIKV infection are hampered by pre-existing cross-reactive antibodies against other flaviviruses[20–22]. Identifying new targets in the ZIKV proteins that are recognized by early antibodies post-exposure and do not cross-react with other flaviviruses could help development of better differential serodiagnostic test for ZIKV infection.

"Whole-Genome-Fragment-Phage-Display-Libraries" (GFPDL) has been previously used for an unbiased comprehensive analysis of the antibody repertoires in individuals infected with viruses, either early or during recovery[23,24]. They can also help to determine the diversity of epitopes bound by post-vaccination sera and decipher the impact of novel adjuvants[25–28]. Such information could help in the development of improved vaccines, therapeutics, and diagnostics. For example, in HIV, panning of virus-specific GFPDLs with sera from acute infections identified several antigenic peptides for differential serodiagnosis of HIV infection from vaccination[29,30], and in avian H5N1, identified peptides for serodiagnosis across multiple clades[31,29].

In the current study, GFPDL spanning the entire genome of ZIKV was constructed and used for in-depth immune profiling of IgG and IgM antibody repertoires in both serum and urine body fluids from individuals acutely infected with ZIKV. We also evaluated total binding and affinity maturation of antibodies against ZIKV NS1 and E-proteins and their evolution during the first month post ZIKV infection using surface plasmon resonance (SPR). The results showed unlinked evolution of antibody responses in terms of antibody epitope repertoire and affinity maturation against structural and non-structural proteins following ZIKV infection in humans, suggesting differential recognition of various ZIKV proteins by the human immune system.

## Results

**Study samples.** We analyzed blood and urine samples from 19 patients (10 females and 9 males; 18–51 years old) with confirmed acute ZIKV infection in Mexico (Supplementary Table 1). Patients were enrolled in the observational cohort between 30 August and 3 November 2016. Only 1/19 individual reported a known prior exposure to Dengue virus (DENV). Both serum and urine samples from all individuals at all time points of collection were PCR negative for DENV infection. All these serum samples (and corresponding urine samples) were from acutely ZIKV-infected adults collected within 0–5 days of onset of symptoms that were PCR positive for ZIKV RNA in serum/urine (Supplementary Table 1). Of the 19 individuals, 11 were PCR positive for ZIKV RNA in both serum and urine, 2 were ZIKV positive only in serum, whereas 6 were only ZIKV positive in urine within the first 7 days of visit (day 0–day 12 since onset of symptoms). The number of clinical symptoms following ZIKV infection in these adults were highest at day 0 visit, and declined by day 28, in most patients (Supplementary Table 2). For simplicity, samples are referred by the visit day throughout the article rather than days post onset of symptoms. For most individuals, the first visit ranged between 0 and 5 days from the day of symptom onset.

**Affinity selection of ZIKV-GFPDL with serum and urine samples.** Whole-genome ZIKV-GFPDL was constructed from the entire genome of ZIKV strain Paraiba_01/2015 (Supplementary Fig. 1). Sequencing of the ZIKV-GFPDL clones showed unbiased random distribution of peptides with diversity in size including large inserts (>500 bp) that spanned the entire ZIKV genome (Supplementary Fig. 2).

To ascertain that the GFPDL represents both linear and conformational epitopes, we performed three independent experiments. First, the ZIKV-GFPDL was used to map epitopes of a panel of linear and conformation-dependent MAbs. ZV54 is ZIKV-specific neutralizing MAb. Structurally, it binds the lateral ridge in DIII of the envelope protein similar to MAb ZV67[32] (Supplementary Fig. 3). MAb ZV67 is a cross-reactive neutralizing and protective mouse MAb recognizing conformational epitope in the lateral ridge of E-domain III[32] (Supplementary Fig. 4). MAb Z23 is DENV-negative, ZIKV-specific neutralizing and protective human MAb that recognizes a conformation-dependent tertiary epitope in E-domain III; mainly binds to one envelope protein monomer and can interact with two envelope protein dimers on the virion surface[15] (Supplementary Fig. 5), MAb ZKA64 (neutralizing and protective human MAb that recognizes a conformational epitope in E-domain III)[14] (Supplementary Fig. 6).

For all four MAbs, strong binding to the ZIKV-GFPDL was observed (Supplementary Figs. 3–6). Importantly, the consensus epitope sequences obtained through GFPDL analysis were very similar to the footprints previously identified for these MAbs[14,15,32]. The reactivity was confirmed by phage - enzyme linked immunosorbent assay (ELISA) using three phages expressing overlapping sequences for each of the MAbs (Supplementary Figs. 3–6, panel B). These results provided proof of concept that ZIKV-GFPDL approach can identify conformational epitopes recognized by previously described protective ZIKV-E-specific MAbs.

Second, we determined the capacity of the phage display library to adsorb ZIKV-E-specific antibodies in the post-ZIKV-infected polyclonal human sera and urine. After two rounds of adsorption with the ZIKV-GFPDL, pooled day 7 serum or urine samples demonstrated >90% removal of total anti-ZIKV-E binding antibodies as measured by SPR (Supplementary Fig. 7). Third, reactivity of the GFPDL-adsorbed sera was evaluated against the ZIKV particles in ELISA, demonstrating >90% adsorption of serum antibodies by the ZIKV-GFPDL (Supplementary Fig. 8). Together, these preliminary studies provided strong rationale of using this GFPDL for epitope mapping of post-ZIKV infection polyclonal sera/urine antibody repertoire.

The ZIKV-GFPDL was used for comprehensive analyses of antibody repertoires of blood and urine samples from 5/19 individuals with confirmed acute ZIKV infections in Mexico. For GFPDL analysis, serum (and corresponding urine samples) were pooled from five acutely ZIKV-infected patients (highlighted in Supplementary Table 1). Subjects 41-010-F, 41-017-F, 42-001-F, 42-003-F, and 42-018-F) gave PCR-positive results for ZIKV RNA in serum and/or urine on visit 1. They were 1–3 days post onset of symptoms. In ELISA, these patients had low anti-ZIKV reactivity of IgM (optical density (O.D.) < 0.1) and IgG (O.D. < 0.4) antibodies that increased on subsequent visits. In all cases, low to moderate reactivity was also observed in DENV ELISA, confirming the known cross-reactivity of antibodies between these closely related flaviviruses (Supplementary Table 1). While none of these five subjects reported prior DENV infection, we cannot exclude the possibility of prior DENV exposure based on the observed ELISA binding results and the high dengue seroprevalence reported in the region[33].

As a negative control, we used a serum from flavivirus naive individual. This serum bound very few phages of the ZIKV-GFPDL (412 and 103 phages bound by IgM and IgG antibodies, respectively) (Fig. 1a). Sequencing of these bound phage clones showed random distribution across the entire ZIKV genome both for IgM and IgG antibody profile (Fig. 1b, c marked "Naive serum").

The pooled sera (day 0 vs. day 7) and urine samples (day 7) from the five acutely infected individuals were subjected to panning with the ZIKV-GFPDL (Fig. 1). Both the IgM and IgG antibody epitope repertoires were evaluated. Higher total numbers of bound phages by serum IgM than IgG antibodies (2–3 logs) were observed on day 0 (first visit) (Fig. 1a). An increase of 1.5–2 log in the numbers of bound phages was observed on day 7 compared with day 0 for both IgM and IgG, confirming the acute infection status of the study participants (Fig. 1a). Urine samples at day 7 contained IgM that bound large numbers of ZIKV-GFPDL phages. However, very few IgG-bound phages were isolated suggesting predominantly ZIKV-specific IgM antibodies in the urine (Fig. 1a).

The inserts of the bound phages were sequenced and mapped against the ZIKV genome (Fig. 1b). Overall, 36 antigenic sites were recognized by the IgM antibodies in the serum and urine of the acutely infected individuals (Fig. 2a). At study day 0, serum IgM antibodies recognized a very diverse array of epitopes spanning the entire ZIKV genome except for NS2A. The capsid region was only minimally recognized (multiple phages bound but with frequency of 1 for each unique peptide). The relative frequencies of bound phages expressing different antigenic sites are presented in Supplementary Table 3. By day 7, a significant immune focusing was observed for IgM antibodies with increased binding to antigenic sites in the E (site Z-6; aa 484–535; 8%), NS3 (site Z-19; aa 1792–1877; 14%), and NS5 (site Z-33; aa 3194–3168; 9% and site Z-35; aa 3308–3368; 6%) (Figs. 1b, 2b). The distribution of epitopes recognized by urine IgM antibodies (day 7 visit) was similar to the binding pattern of the serum IgM antibodies with predominant binding to the antigenic sites in NS3, but not similar in NS5 (Figs. 1b, 2b and Supplementary Table 3).

The IgG response in sera on day 0 post onset was much more limited compared with the IgM response (2 logs fewer bound phages). The IgG antibodies showed binding to diverse antigenic sites mostly in the structural proteins (prM/E) and non-structural proteins NS3 and NS5, but no immunodominant (clonal frequency >10%) IgG antigenic site (Figs. 1c and 2c). However, by day 7, the IgG antibodies in the serum from the same individuals demonstrated pronounced immune focusing to several antigenic sites in E (Z-8; aa 595–729; 16%), NS1 (Z-14; aa 1046–1127; 7%), and NS2B (Z-15; aa 1417–1474; 49%) (Figs. 1c, 2c and Supplementary Table 3). These data suggested early expansion of IgM and isotype-switched IgG B cells recognizing different ZIKV proteins following acute ZIKV infection. Interestingly, minimal ZIKV-specific IgG antibodies were observed in the urine samples from the same individuals at day 7. Subsequently, additional IgM and IgG antibody epitope repertoire analysis was performed with serum sample of an acutely ZIKV-infected individual (patient # 42–001-F) at day 7 visit. This individual was part of the five pooled samples used for GFPDL analysis in Figs. 1 and 2. The epitopes recognized by IgM and IgG antibodies identified similar pattern to the pooled samples (Supplementary Fig. 9). Again, the IgG-bound epitopes were more focused than the IgM repertoire, with immunodominance of NS2B and E.

**Surface exposure of antigenic sites identified by ZIKV-GFPDL.** It was important to determine whether the epitopes recognized by the ZIKV post-infection serum and urine antibodies, as identified by the GFPDL panning, are likely to be exposed on the surface of

the ZIKV proteins. This was done for available protein structures: mature and immature E (Fig. 3), NS1 (Supplementary Fig. 10), NS2B (Supplementary Fig. 11), NS3 (Supplementary Fig. 12), and NS5 (Supplementary Fig. 13). In addition to surface mapping of the antigenic sites on the protein structures, heat maps showing the protein sequence conservation for each antigenic site, either among ZIKV strains or between ZIKV and other flaviviruses are presented (panel a in Fig. 3 and Supplementary Figs. 10–13). The majority of antigenic sites identified by GFPDL were expressed on the surface of the individual proteins, whereas some antigenic sites were only partially exposed (panel b in Fig. 3 and Supplementary Figs. 10–13). The GFPDL analyses identified many antigenic sites spanning prM-E (Figs. 1 and 2 and Supplementary Table 3). These epitopes were mapped on the crystal structures of ZIKV immature or mature forms of the prM and E protein (Fig. 3b).

**Conservation of antigenic sites in ZIKV strains and flaviviruses.** Prior exposure to other flaviviruses may play a role in the observed antibody responses to ZIKV infection. We plotted the % similarity among sequences of old and new ZIKV strains (Supplementary Fig. 11a), and % similarity between flaviviruses (DENV 1–4, Yellow Fever virus, and West Nile virus) (Supplementary Fig. 11b). In both cases, the sequence of ZIKV_Paraiba strain was used as a reference sequence (considered as 100%). In general, the ZIKV sequences from strains isolated between 1947 and 2015 in diverse geographical areas are highly conserved (Supplementary Fig. 14a), but conservation between ZIKV and other flaviviruses (DENV 1–4, West Nile virus, and Yellow Fever virus) is highly variable (Fig. 3a, Supplementary Fig. 14b and Supplementary Table 4). Within the ZIKV-E protein, the GFPDL analysis identified IgM and IgG binding to multiple domains including Domain III (Z-8/Z-8.1 in Fig. 2a, Fig. 3b). Domain III of ZIKV is divergent compared with other flavivirus E-proteins (Supplementary Fig. 12). Among ZIKV strains, sites of modest variability were identified in the prM, and domains I and II of the E protein covered by antigenic sites Z-3 to Z-6 in the GFPDL analysis, whereas E-Domain III antigenic sites (Z-8, Z-8.1, and Z-9) and NS1 antigenic sites (Z-10 to Z-14) are highly conserved (Supplementary Fig. 14a). Most of the antigenic sites recognized by the IgM and IgG antibodies in the serum or urine post-ZIKV infection that mapped to the ZIKV non-structural proteins were highly conserved within the ZIKV strains (Supplementary Fig. 14a). However, different levels of sequence conservation were observed with other flaviviruses (Z-10 to Z-36) (Supplementary Table 4). The NS3 contains a single site of variability, which was not covered by any of the antigenic sites identified by either the IgM or IgG antibodies in the GFPDL analysis (Fig. 2a and Supplementary Fig 14a). When comparing ZIKV sequence with all other flaviviruses, much larger sequence diversity scores were found in regions spanning E-domain I (Z-5), E-domain III (Z-8, Z-9), NS1 (Z-10 and Z-11), NS2B (Z-15), NS4A (Z-22), NS4B (Z-23–24), and NS5 (Z-28) (Supplementary Table 4, Supplementary Figs. 14b and 15). The Yellow Fever viral genome was the most divergent of all the flaviviruses. Therefore, the antigenic sites identified by post-infection serum (IgM/IgG) and urine (IgM) spanned both sites of high conservation and several sites of divergence compared with other flaviviruses. Several sequences in the non-structural highly conserved ZIKV proteins (NS1, NS2B, NS3, NS4B, and NS5), recognized at high frequencies by the IgM and IgG antibodies in the serum or urine post-ZIKV infection, could potentially be used for serodiagnosis of ZIKV infection (Supplementary Fig. 14 and Supplementary Table 4).

In summary, the GFPDL analysis revealed highly diverse antibody repertoires in serum and urine samples from ZIKV-infected

**a**

| | IgM antibodies | | | | IgG antibodies | | | |
|---|---|---|---|---|---|---|---|---|
| | Naive serum | Serum D0 | Serum D7 | Urine D7 | Naive serum | Serum D0 | Serum D7 | Urine D7 |
| Phage titer | 412 | $8.5 \times 10^7$ | $2.02 \times 10^8$ | $2.22 \times 10^8$ | 91 | $9.7 \times 10^4$ | $5.71 \times 10^6$ | 103 |

**Fig. 1** IgG and IgM repertoires across Zika virus (ZIKV) proteome following Zika virus infection. **a** Number of bound phage clones isolated using ZIKV GFPDL affinity selection against flavivirus naive negative control serum, pooled serum samples from five acutely ZIKV-infected patients on day 0 and day 7 of hospital visit, and urine sample on day 7 of hospital visit (day 0 of hospital visit corresponds to 0–3 days since onset of symptoms for these five patients). **b**, **c** Schematic alignment of the peptide sequences recognized by IgM (**b**) and IgG (**c**) antibodies in ZIKV-infected human sera (day 0 and day 7) and urine (day 7) and flavivirus naive negative control serum, identified by panning with ZIKV-GFPDL. The amino-acid designation is based on the ZIKV polyprotein sequence encoded by the complete *ZIKV-ICD* genome (Supplementary Fig. 1). Bars indicate identified epitopes in the different structural (C, prM, E) and non-structural (NS) genes on the ZIKV polyprotein sequence. Graphical distribution of representative clones with a frequency of >2, obtained after affinity selection, are shown. The horizontal position and the length of the bars indicate the peptide sequence displayed on the selected phage clone to its homologous sequence in the ZIKV sequence on alignment. The thickness of each bar represents the frequency of repetitively isolated phages. Phage sequences and clonal frequency is shown in Supplementary Table 3

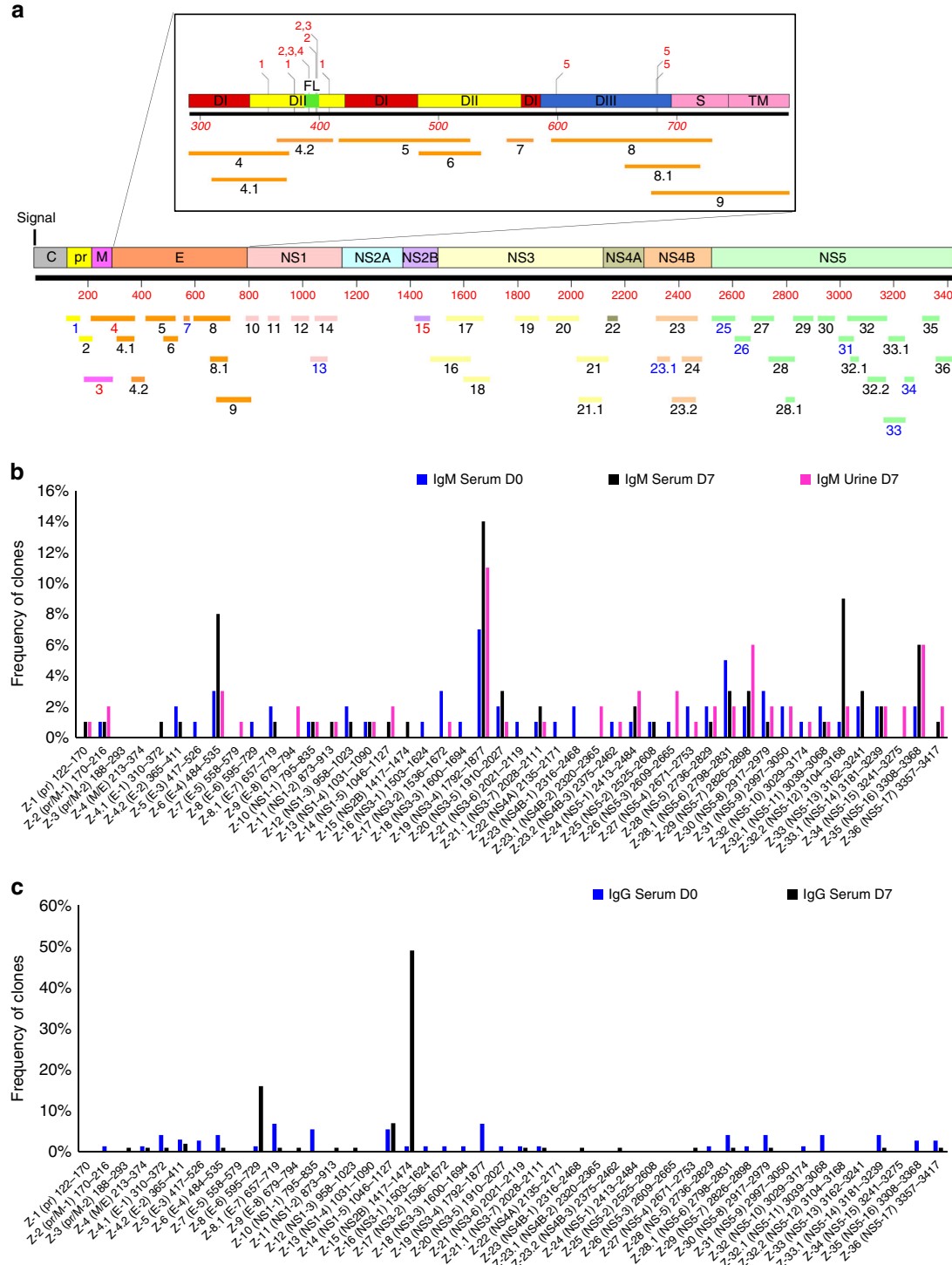

**Fig. 2** Frequency of Zika virus (ZIKV) epitopes in serum and urine following Zika virus infection. **a** Antigenic sites within the ZIKV proteins recognized by serum (days 0, 7) and urine (day 7) IgM and IgG antibodies following Zika virus infection (based on data presented in Fig. 1). The amino-acid designation is based on the ZIKV polyprotein sequence encoded by the complete ZIKV genome (Supplementary Fig. 1). The amino-acid designations on the ZIKV polyprotein (Supplementary Fig. 1) are as follows: C capsid, pr peptide pr, M membrane, E envelope protein, NS non-structural protein. Inset shows expanded version of E protein schematic with domains (D) I, II, and III and fusion loop (FL) shown along with their antigenic sites. Previously described epitopes using MAbs are shown above the ZIKV-E schematic. Critical residues for binding of MAbs, 1, ZIKV-117; 2, 2A10G6; 3, ZIKV-12; 4, ZIKV-15; and 5, ZIKV-116 are depicted. **b**, **c** Distribution and frequency of phage clones expressing each of the key ZIKV antigenic sites recognized by IgM (**b**) and IgG (**c**) antibodies in post-ZIKV infection serum (day 0 and day 7) and urine (day 7) are shown. The number of phage clones that expressed each antigenic site was divided by the total number of ZIKV-GFPDL selected clones for each pooled sera or urine samples and represented as a percentage. Phage sequences and clonal frequency is shown in Supplementary Table 3

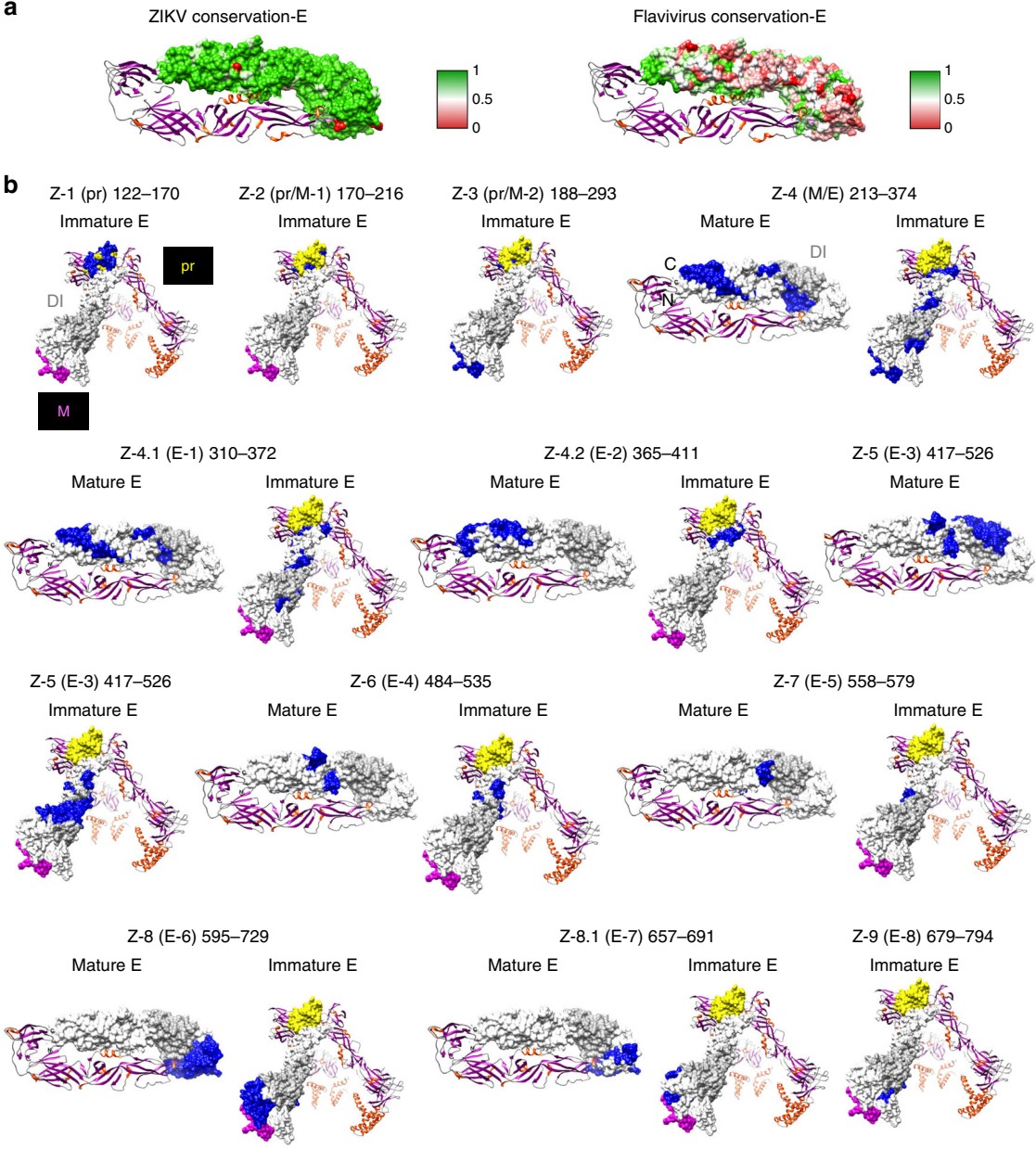

**Fig. 3** Structural representation of antigenic sites identified in ZIKV-E protein. **a** (Left panel) Heat map showing sequence conservation on one monomer chain of mature ZIKV-E protein structure (PDB# 5JHM) based on comparison with several Zika virus (ZIKV) isolates (Paraiba, Uganda-1947, Nigeria-1968, Senegal-2001, Micronesia-2007, and Brazil-2016 strains) and (right panel) ZIKV vs. other flaviviruses (Dengue virus (DENV) 1 to 4, West Nile virus, Yellow Fever virus and ZIKV). The heat map has been color coded from red (0) to green (1), where green denotes complete conservation. **b** Antigenic sites within ZIKV-prM/E identified by the GFPDL analysis are depicted in blue on the structures of both immature (PDB 5U4W) and mature ZIKV-E (PDB 5JHM). Domain I is shaded in light gray (PDB 5JHM), pr domain is shaded in yellow and M in pink (PDB 5U4W). PDB Structure# 5U4W encompasses residues 288–794 and PDB structure# 5JHM encompasses residues 313–699 based on ZIKV_ICD polyprotein sequence (Supplementary Fig. 1)

individuals that bound to multiple non-structural genes (except NS2A) in addition to prM/E. Differential antibody binding profiles were observed for IgM and IgG antibodies following ZIKV infection and immune focusing of the antibody repertoire was observed on day 7 compared with day 0.

For further evaluation of diagnostic potential of the identified ZIKV antigenic sites, representative peptides spanning selected antigenic sites were synthesized and tested with individual acute ZIKV infected, convalescent ZIKV, and convalescent DENV-infected serum samples in ELISA (Fig. 4). Interestingly, most acute ZIKV-infected sera reacted with NS2B peptide (Z-15) at least once, either at day 0 or day 28 visit (17/19 were seropositive

at least once on either day; 6/19 and 15/19 with end-point titers >1:00, respectively). NS2B peptide also showed high reactivity (13/13) with all ZIKV convalescent sera in ELISA (Fig. 4a, b). Importantly, only 1/27 convalescent sera from DENV-infected individuals reacted weakly (end-point tier of 100) with the NS2B peptide, demonstrating ZIKV antibody specificity for this peptide sequence. Peptides from NS1 (1033–1067 within Z-13), NS3 (1805–1873 within Z-19), ZIKV-NS4B (2422–2465 within Z-24), ZIKV-NS5 (2860–2901 representing Z-29), showed variable reactivity (50–76%) with the acutely ZIKV-infected samples, moderate reactivity to ZIKV convalescent samples, and minimal or no reactivity to DENV-infected samples (Fig. 4a, b).

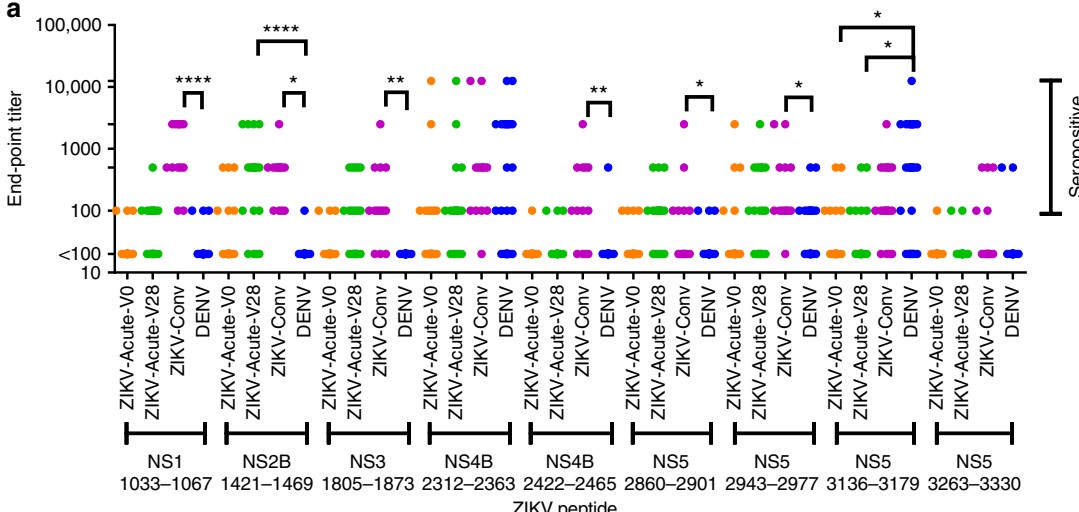

**b**

| Peptide | Sample | Frequency of values ≥100 | Percent of values ≥100 (%) |
|---|---|---|---|
| ZIKV NS1 (1033–1067) | ZIKV-Acute-V0 | 3/19 | 15.79 |
| | ZIKV-Acute-V28 | 9/19 | 47.37 |
| | ZIKV-Conv | 13/13 | 100.00 |
| | DENV | 3/27 | 11.11 |
| ZIKV NS2B (1421–1469) | ZIKV-Acute-V0 | 6/19 | 31.58 |
| | ZIKV-Acute-V28 | 15/19 | 78.95 |
| | ZIKV-Conv | 13/13 | 100.00 |
| | DENV | 1/27 | 3.70 |
| ZIKV NS3 (1805–1873) | ZIKV-Acute-V0 | 3/19 | 15.79 |
| | ZIKV-Acute-V28 | 11/19 | 57.89 |
| | ZIKV-Conv | 10/13 | 76.92 |
| | DENV | 0/27 | 0.00 |
| ZIKV NS4B (2312–2363) | ZIKV-Acute-V0 | 8/19 | 42.11 |
| | ZIKV-Acute-V28 | 13/19 | 68.42 |
| | ZIKV-Conv | 12/13 | 92.31 |
| | DENV | 15/27 | 55.56 |
| ZIKV NS4B (2422–2465) | ZIKV-Acute-V0 | 1/19 | 5.26 |
| | ZIKV-Acute-V28 | 3/19 | 15.79 |
| | ZIKV-Conv | 9/13 | 69.23 |
| | DENV | 1/27 | 3.70 |
| ZIKV NS5 (2860–2901) | ZIKV-Acute-V0 | 4/19 | 21.05 |
| | ZIKV-Acute-V28 | 14/19 | 73.68 |
| | ZIKV-Conv | 6/13 | 46.15 |
| | DENV | 3/27 | 11.11 |
| ZIKV NS5 (2943–2977) | ZIKV-Acute-V0 | 5/19 | 26.32 |
| | ZIKV-Acute-V28 | 12/19 | 63.16 |
| | ZIKV-Conv | 12/13 | 92.31 |
| | DENV | 12/27 | 44.44 |
| ZIKV NS5 (3136–3179) | ZIKV-Acute-V0 | 6/19 | 31.58 |
| | ZIKV-Acute-V28 | 15/19 | 78.95 |
| | ZIKV-Conv | 12/13 | 92.31 |
| | DENV | 20/27 | 74.07 |
| ZIKV NS5 (3263–3330) | ZIKV-Acute-V0 | 1/19 | 5.26 |
| | ZIKV-Acute-V28 | 2/19 | 10.53 |
| | ZIKV-Conv | 5/13 | 38.46 |
| | DENV | 2/27 | 7.41 |

The ZIKV-NS5-3263-3330 peptide showed weak reactivity to few (7–38%) convalescent ZIKV and DENV-infected samples. The peptides ZIKV-NS4B-2312-2363 (overlapping Z-23.1), ZIKV-NS5-2943-2977 (within Z-30) and ZIKV-NS5-3136-3179 (overlapping Z-32.2) all showed weak/moderate reactivity with acute ZIKV-infected samples (68–76%), but similar reactivity to convalescent sera from ZIKV and DENV-infected individuals (60–92%), reflecting the higher degree of antigenic conservation in these peptide sequences among these flaviviruses (Fig. 4a, b).

**Fig. 4** Seroreactivity of Zika virus (ZIKV)- and Dengue virus (DENV)-infected samples with ZIKV peptides. **a** End-point titers of various acute (Acute-V0; orange symbols and Acute-V28; green symbols) and convalescent ZIKV (ZIKV-Conv; magenta symbols) and convalescent DENV human serum samples (DENV; blue symbols) tested for binding to various ZIKV antigenic site peptides in ELISA are depicted. ELISA was performed with fivefold serially diluted (starting at 1:100) samples. One-way ANOVA was performed with a Bonferroni post-hoc analysis. ****$p < 0.0001$, **$p < 0.005$ and *$p < 0.05$. **b** Table showing frequency and percentage of samples with end-point titers of ≥100 for each of the sample groups; ZIKV-Acute-V0 and 28, ZIKV-Conv and DENV for each ZIKV peptide determined by peptide ELISA. Frequency and percentage seropositivity were calculated for 19 samples each of ZIKV-Acute-V0 and V28, 13 samples for ZIKV-Conv, and 27 samples for convalescent DENV-infected samples

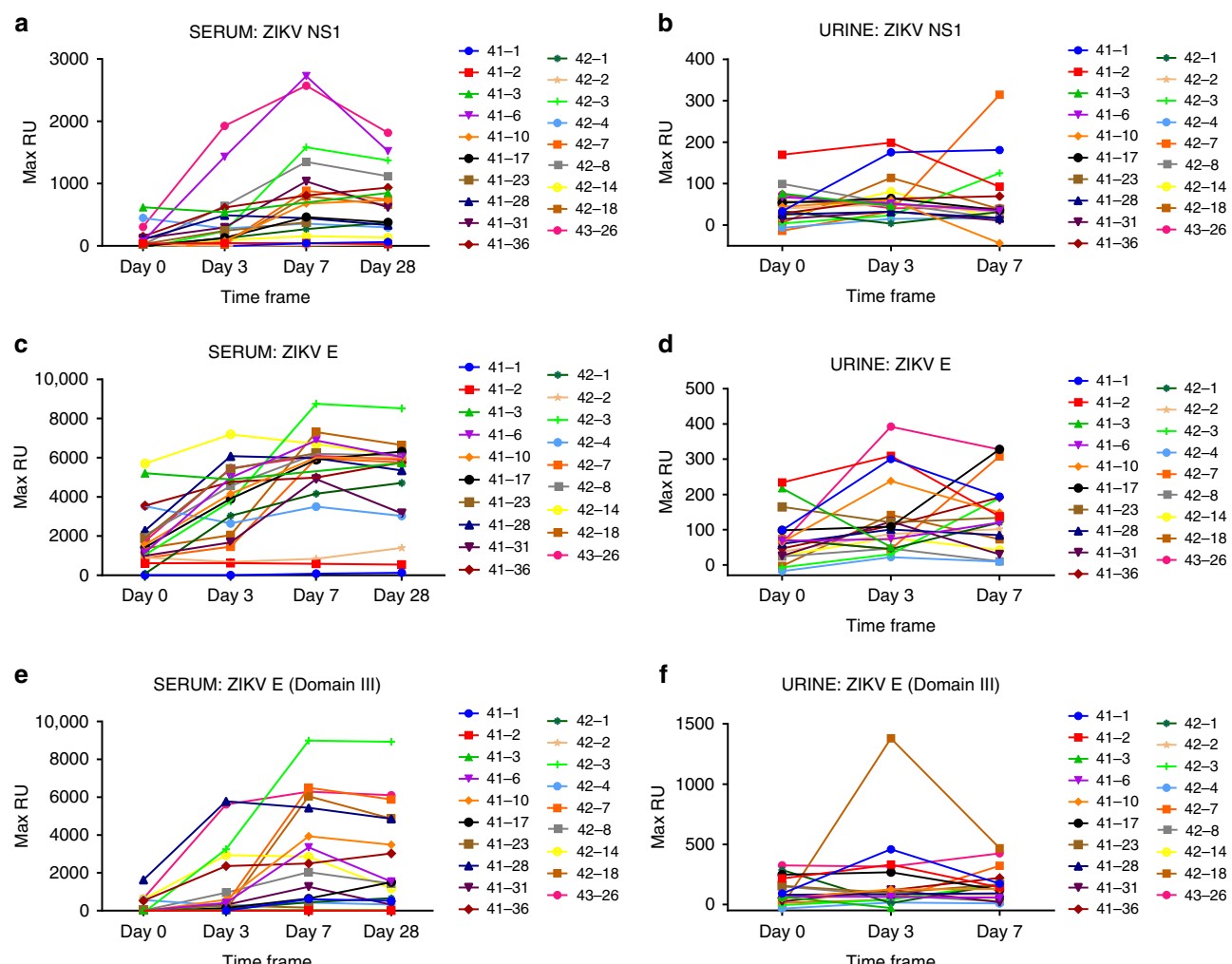

**Fig. 5** Binding of Zika virus (ZIKV)-infected human serum and urine to NS1, E, and E-dIII in SPR. Serum (**a**, **c**, **d**) and urine (**b**, **d**, **f**) samples collected at different time points from adults post-ZIKV infection (days 0, 3, 7, and 28) were analyzed for total binding to purified NS1 (**a**, **b**), ZIKV-E (**c**, **d**), and ZIKV-E-domain III (**e**, **f**) proteins. Total antibody binding is represented in SPR resonance units. Maximum resonance unit (Max RU) values for protein binding by serum or urine antibodies obtained from each individual at different time point post-ZIKV exposure are linked by connecting lines. Source data are provided as a Source Data file

**Antibody binding kinetics to ZIKV-E, domain III, and NS1.** The evolution of antibody binding kinetics in serum and urine samples of each ZIKV-infected individual against the structural and non-structural proteins of ZIKV was performed using SPR-based real-time kinetics assay. The ZIKV Envelope (E) protein is an important target for vaccine development, whereas NS1 is important for ZIKV diagnosis[34]. Therefore, ZIKV-E-ectodomain, E-domain III, and NS1 proteins were used for the SPR analysis of serum and urine samples from day 0, day 3, day 7 and day 28 visits from ZIKV-infected individuals (Fig. 5). Serum antibody binding to NS1 was variable. For the majority of patients, the total binding maximum resonance unit (Max RU) started to increase on day 3, peaked on day 7, and either plateaued or started to drop

by day 28 visit (Fig. 5a). However, 11% (2/19) of ZIKV-infected individuals did not show measurable serum antibody binding (Max RU < 20) to NS1 protein following ZIKV exposure. Urine samples from all subjects showed very low NS1 antibody binding on all days, with only four post-infection samples binding (Max RU > 20) on day 3 or 7 (Fig. 5b).

In contrast to NS1, serum antibody binding to the E-ectodomain and to E-domain III was more robust following ZIKV infection, peaking either on day 3 or day 7 visits (day 3 to day 12 post onset of symptoms) and were largely maintained until day 28 with a moderate decline in few individuals (Fig. 5c, e). The total serum antibody binding to either E-ectodomain or Domain III was about threefold higher compared with the antibody

binding to NS1. However, 16% (3/19) of ZIKV-infected individuals did not show measurable serum antibody binding (Max RU < 20) to either the E or domain III proteins following ZIKV exposure. Two of the three ZIKV-E non-responsive individuals were the same individuals that did not show reactivity to NS1 protein (samples 41-2 and 42-2), and the third individual had very low NS1 binding (sample 41-3). Interestingly, these three individuals gave positive reactivity in the commercial ZIKV-ELISA at the first visit but did not show increase in either IgG or IgM binding titers on subsequent visits (Supplementary Table 1). Only one of these individuals (42-2) acknowledged prior DENV infection (Supplementary Table 1). However, we cannot exclude the possibility that these individuals developed some antibodies to quaternary epitopes in ZIKV-E, E-domain III, or NS1 that were not captured by the SPR-binding assay.

Antibody binding of urine samples to E-proteins was generally much lower compared with the serum antibody binding, and there was no direct correlation between reactivity of serum and urine antibodies in SPR. These data suggested that not all ZIKV infections lead to seroconversion, which could lead to underestimation of exposure rates in antibody-based serodiagnostic tests.

To further evaluate antibody affinity maturation following ZIKV infection, the antibody–antigen complex dissociation rates (off-rate constants) were determined as a surrogate for affinity maturation using SPR. Antibody dissociation kinetics of antigen–antibody complexes are independent of antibody concentration and provide a measure of overall average affinity of polyclonal antibody binding. To that end, serially diluted serum/urine at 10- and 40-fold dilutions were injected (120-s contact time) for association, and dissociation was performed over a 600-s interval on ZIKV proteins (Supplementary Fig. 16a, b, respectively). Antibody off-rate constants, which describe the fraction of antigen–antibody complexes that decay per second, were determined directly from the plasma antibody interaction with E or NS1 proteins in the dissociation phase only for the sensorgrams with maximum RU in range of 10–100 RU using BioRad Proteon SPR machine (Supplementary Fig. 16)[25]. Antibody binding to NS1 showed low affinity with fast dissociation rate kinetics (off-rates of $10^{-1}$ to $10^{-2}$/s). Furthermore, only minimal affinity maturation against NS1 was observed between day 0 and day 28 post infection (Fig. 6a). In contrast, antibody binding to the E-ectodomain and E-Domain III, demonstrated a significant ($p < 0.05$; two-tailed paired $t$-test) affinity maturation

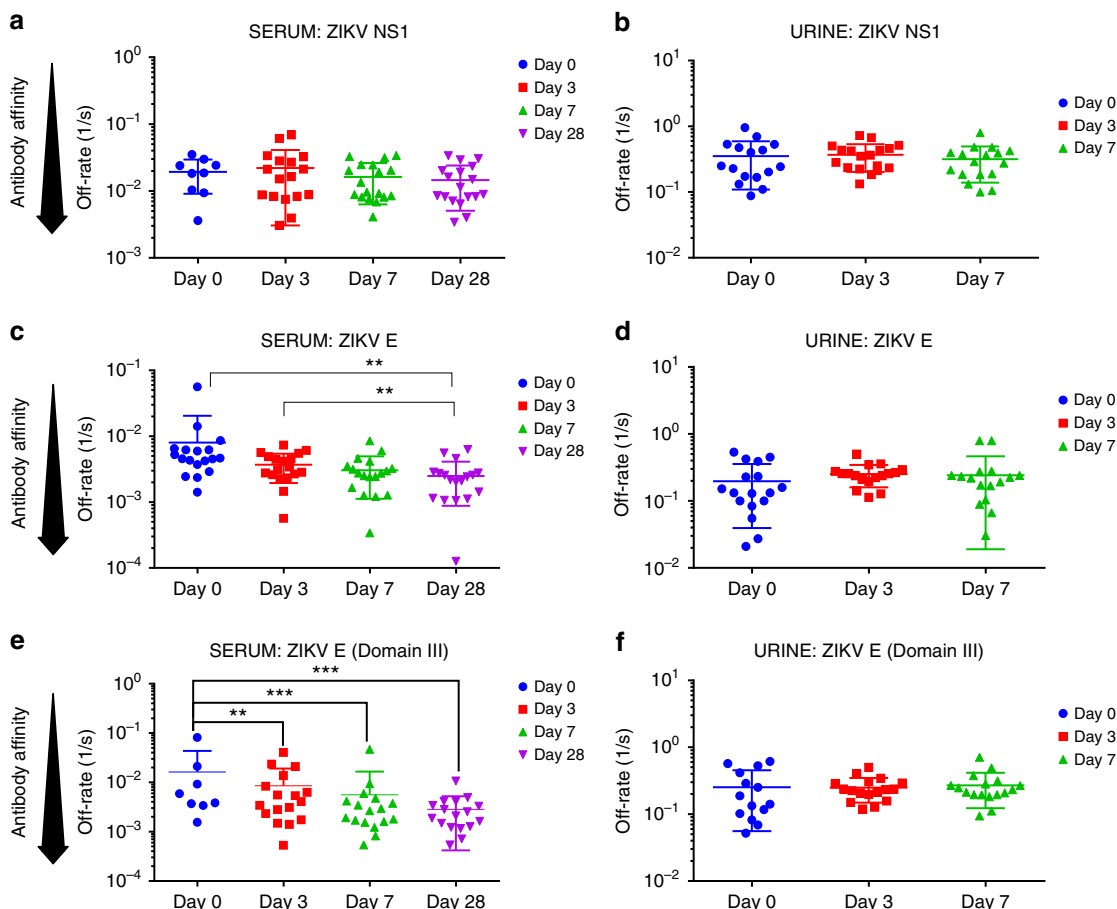

**Fig. 6** Antibody affinity maturation in Zika virus (ZIKV)-infected serum and urine to NS1, E, E-dIII. SPR analysis of individual sera (**a**, **c**, **e**) or urine (**b**, **d**, **f**) post-ZIKV infection was performed with purified recombinant NS1 (**a**, **b**), ZIKV-E (**c**, **d**), and ZIKV-E-domain III (**e**, **f**) proteins to determine the dissociation kinetics (off-rates) at different time points post infection. Antibody off-rate constants that describe the fraction of antibody–antigen complexes decaying per second were determined directly from the serum/urine sample interaction with recombinant and NS1, ZIKV-E, ZIKV-E-domain III proteins using SPR in the dissociation phase as described in Methods. Mean values with standard deviation are shown by bar for each group. Antibody off-rates were not calculated for samples with total antibody binding (Max RU shown in Fig. 5) of <10 at 10-fold dilution and is not shown in figure. The statistical significances between each time point (visit days) were determined using two-tailed paired $t$-test in GraphPad software. $p$-Values <0.05 were considered significant with a 95% confidence interval. Statistically significant with $p$-values of <0.005 (**), or <0.001 (***) are shown. Source data are provided as a Source Data file

of serum antibodies between days 0 and 28 (Fig. 6c, e), with off-rates ranging between $10^{-2}$ and $10^{-3}$/s. The antibody affinity against the E-ectodomain and domain III was ~10-fold higher compared with the NS1 protein. No antibody affinity maturation was observed for urine antibodies (Fig. 6b, d, f), which is in line with the fact that only IgM antibodies were found in the urine samples (Fig. 1).

To determine the relevance of antibody affinity maturation with the clinical disease following ZIKV infection, Spearman correlations were determined for the polyclonal sera antibody off-rates to ZIKV-E protein and NS1 proteins on the day of peak antibody binding titer (day 7) vs. number of clinical symptoms on day 28 visit. The first visit date ranged from 0 to 5 days post onset of symptoms, and onset of symptoms can be 3–12 days after infection. While this is still an acute infection response with predominant IgM responses, there was evidence of class switching and affinity maturation between visit day 0 and visit day 7. Therefore, it was reasonable to probe the correlation between early antibody affinity maturation and the reduction in clinical symptoms by day 28. Statistically significant inverse correlations were observed between polyclonal serum antibody off-rates to ZIKV-E and the number of clinical symptoms on day 28 ($p = 0.01569$; Spearman test; Fig. 7a). In contrast, no significant correlation was found between binding serum antibody affinities to the NS1 protein and number of symptoms on day 28 (Fig. 7b).

Together, our data demonstrate differential evolution of antibody diversity, class switching and affinity maturation within structural and non-structural proteins in different body fluids following ZIKV exposure. Some of the unique antigenic sites identified in our study could be further evaluated as serodiagnostic targets, as well as countermeasures against ZIKV disease. Antibody affinity maturation against ZIKV-E protein may play an important role in resolution of clinical symptoms following ZIKV infection.

## Discussion

Protection against ZIKV disease is at least partially attributed to the humoral immune response, since strong correlation was demonstrated between ZIKV-specific antibody responses and protective efficacy after vaccination of mice and non-human primates (NHPs). Furthermore, passive transfer of antibodies to naive NHPs can protect the recipients against ZIKV challenge[35–37].

However, there is limited knowledge of the specificities of the antibodies generated following ZIKV infection in humans and their evolution over time. To address this need, in the current study, we used unbiased technologies including ZIKV-GFPDL

and SPR to perform a comprehensive analysis of the evolution of antibody repertoires across the whole-viral proteome in a group of patients with confirmed exposure to the ZIKV.

The ZIKV-GFPDL panning identified very broad antibody reactivities as early as day 0 (first visit: 0–3 days post symptoms onset) that were predominantly IgM antibodies. The only region not recognized by serum antibodies was the NS2A protein, even though it was well represented in the ZIKV-GFPDL (Supplementary Figs. 1 and 2). One of the possible limitations of GFPDL-based assessments is that they are unlikely to detect paratopic interactions that require post-translational modifications and rare quaternary epitopes formed by ZIKV proteins. Binding to NS4A was also very low (only one insert bound). The ZIKV genome is organized as a single open reading frame transcribed and translated as a single polyprotein, which undergoes proteolytic cleavage by host and viral proteases to generate three structural C, prM/M, and envelope E, and seven non-structural proteins. It is possible that different viral proteins are processed with different efficiency following ZIKV infection. More importantly, it is unknown whether if the individual proteins are being released by infected cells at the same frequency. By day 7, there was an increase in the frequency of IgM-bound phages mapping to E-domain II (aa 484–535), NS3 (aa 1792–1877) and several sites in NS5. These are regions with significant homology to other flaviviruses (DENV, West Nile virus) (Supplementary Fig. 14). In Tapachula, Chiapas, Mexico, where the study was conducted, DENV is common[33], yet only one confirmed prior DENV exposure was reported in the current study (Supplementary Table 1). This individual was not included in the samples for the GFPDL analysis. However, we cannot exclude the possibility of unconfirmed prior exposure to other flaviviruses in the study subjects resulting in recall antibody responses shortly after acute ZIKV infection. The urine day 7 IgM antibodies had similar, but not identical, repertoires to the serum IgM antibodies. The presence of IgM in urine, with no evidence of class switching can be explained by local urogenital ZIKV replication, in agreement with previous reports[38,39]. Recent studies suggested that plasma and urine ZIKV PCR positivity are not linked, supporting local infections in different organs resulting in localized immune responses, in agreement with our findings[40]. Very strong IgG binding was observed to antigenic sites in the E protein, NS1 and NS2B, but not to NS3–NS5 proteins. This is surprising, because all the structural and non-structural proteins are derived from a single ZIKV polyprotein and should be in equivalent amounts following proteolytic cleavage. We previously reported unlinked evolution of antibody binding to subdomains in the influenza hemagglutinin (HA1 vs. HA2) and RSV membrane proteins (F vs. G) over time[23–25,28]. The unlinked antibody evolution against

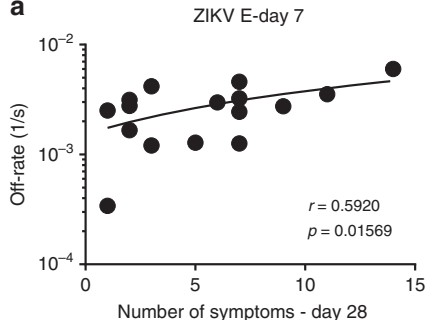
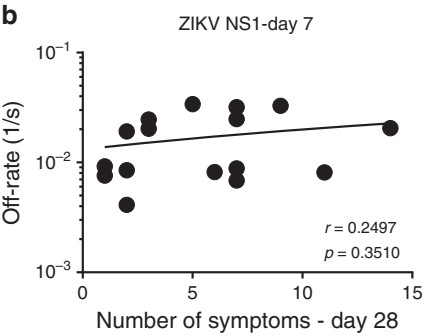

**Fig. 7** Relationship between serum antibody affinity with Zika virus (ZIKV) clinical symptoms. Antibody off-rate constants of the polyclonal serum sample interaction with recombinant ZIKV-E (**a**) and NS1 (**b**) proteins on day 7 visit as measured by SPR was correlated with total number of symptoms on day 28 visit for the corresponding patient. Inverse Spearman correlations were observed between anti-ZIKV-E antibody affinity on day 7 measured by SPR vs. number of symptoms on day 28 ($r = 0.592$; $p = 0.01569$). Source data are provided as a Source Data file

these viral proteins in influenza and RSV is most likely due to multiple mechanisms including differential viral protein expression following virus infection, protein secretion/release from infected cells, antigen presentation, immune dominance, pre-existing immunity, and immune selection over time. In contrast, for flaviviruses, a single polyprotein is being cleaved to generate the structural and non-structural proteins at equimolar amounts. In this study, we measured immune responses against more conserved and less conserved proteins shortly after primary acute ZIKV infection. During that time frame, it is unlikely that prior immunity played a key role in the observed differential antibody responses.

These findings of class switching of IgM antibodies to IgG antibodies and affinity maturation for different matured ZIKV proteins suggest that immune dominance following infection is determined by multiple factors including protein concentration, BCR affinity, antigen processing by B cells, presentation to $T_{FH}$ cells and germinal center (GC) entry, which is required for class switching and affinity maturation[41-44]. The E protein is expressed on both virions and infected cells and was shown to be highly immunogenic in all flaviviruses. NS1 is immunogenic and was shown to contribute to virus pathogenesis in the host, and to enhance viral acquisition by mosquitoes[45-47]. On the other hand, the strong binding of antibodies to NS2B was not described before and merits further investigation.

In order to identify immune correlates of protection, it is important to follow the kinetics of antibody binding and affinity maturation against different viral gene products in individual patients. Interestingly, while most patients had antibodies against NS1, E, and E-domain III during the 28-day observation period (peak responses on day 7), the antibody binding to NS1 was at least threefold lower than the binding of antibodies targeting the E-proteins (Fig. 5). Furthermore, the affinity of anti-NS1 antibodies was 10-fold lower than anti-E antibodies, and very minimal affinity maturation for anti-NS1 antibodies was observed between day 0 and day 28 (Fig. 6). It is highly likely that antibody affinity maturation continues beyond the 28-day period. The difference in IgM and IgG antibody repertoires and affinity maturation suggested that not all the early IgM$^+$ B cells underwent class switching and engaged in GC formation[48]. Alternatively, we cannot exclude the possibility that the serum antibodies may reflect both long term cross-reactive memory B cells and activated naive B cells. Unfortunately, it is not possible to obtain pre-infection serum samples that could help to decipher these possibilities.

Domain III in the E-ectodomain is the least conserved between ZIKV and other arboviruses including dengue (Fig. 3 and Supplementary Fig. 15). Several human MAbs targeting the ZIKV domain III have been shown to have potent virus neutralizing activity[16,32,49]. Our data confirm that polyclonal antibodies from acutely infected patients indeed demonstrate strong binding and affinity maturation against E-Domain III supporting the use of Domain III as a targeted vaccine in naive populations.

The contributions of antibody functions to the control of viremia and clinical symptoms post-ZIKV infection will require continuing studies. Antibody epitope sites, total antibody binding, and antibody affinity maturation may all contribute to epidemiology and disease outcomes. In the current study, we found a statistically significant but modest inverse correlation between the day 7 antibody binding affinity against the E-ectodomain and the number of clinical symptoms on day 28 post-ZIKV infection. The numbers of study participants are limited to draw strong conclusions, but it provides a rationale for measurements of antibody affinity in future vaccine trials, as well as post-exposure studies.

Interestingly, 3/19 of ZIKV PCR-positive individuals in the current study did not show measurable serum antibody binding

to the E, E-domain III, as well as NS1 proteins in SPR. These data suggested that not all ZIKV infections lead to seroconversion, even though they gave positive reactivity in the commercial ZIKV-ELISA on their first visit (with no increase titers in subsequent visits). This could lead to underestimation of exposure rates. A search for additional targets for serodiagnostic and surveillance tests is warranted. Importantly, the NS2B antigenic site (Z-15; Fig. 3) discriminates between Zika and dengue. Furthermore, this site is 100% conserved among ZIKVs, but only 30–55% conserved between ZIKV and other flaviviruses (Supplementary Fig. 14 and Supplementary Table 4). Peptides derived from NS1, NS2B, NS3 and one site in NS4B were highly reactive with both acute and convalescent ZIKV sera (47–79 and 69–100%) with very low reactivity with DENV convalescent sera (0–11% cross-reactivity). Therefore, an optimized combination of NS1, NS2B, NS3, and/or NS4B peptide-based serodiagnostic assay could be developed for surveillance studies.

In summary, our study demonstrated independent evolution of antibody binding patterns to structural and non-structural proteins following acute ZIKV infection in terms of antibody epitope repertoire diversity, antibody affinity maturation, and antibody isotype class switch, including importance of predominant anti-ZIKV IgM response in different body fluids. These findings could have significant implications for further development and evaluation of ZIKV serodiagnostics, therapeutics, and vaccines.

## Methods

**Study design**. We analyzed samples from patients with confirmed ZIKV infection enrolled in a prospective, longitudinal observational study conducted at four hospitals in Tapachula, Chiapas, Mexico: Instituto de Seguridad y Servicios Sociales de los Trabajadores del Estado-Clínica Hospital Dr. Roberto Nettel Flores, Hospital Regional de Alta Especialidad Ciudad Salud, Hospital General de Tapachula, Instituto Mexicano del Seguro Social—Unidad de Medicina Familiar No.11 to study the natural history of Zika. The study was sponsored by the Mexican Emerging Infectious Disease Clinical Research Network, Mexico (La Red), and conducted in accordance with the applicable regulatory and International Conference on Harmonization—Good Clinical Practice requirements. The study protocol was approved by an institutional review board for each study site, as well as by all local and/or country governing bodies as applicable (ClinicalTrials.gov Identifier: NCT02831699).

Individuals of all ages and any gender that met the 12 February 2016 WHO/PAHO case definition (World Health Organization. ZIKV disease: Interim case definitions; http://apps.who.int/iris/handle/10665/204381) of suspected acute Zika (fever and/or rash, and one or more other symptoms including arthralgia, myalgia, non-purulent conjunctivitis or conjunctival hyperemia, headache, and malaise) were eligible for the study if symptoms started in the 6 previous days before first visit. Informed consent was obtained from every participant in this study. After consent, subjects were evaluated on study days 0, 3, 7, and 28 with a series of clinical assessments, and serum and urine samples were obtained at each visit. For this analysis, only subjects with confirmed Zika (PCR positive)[20] in serum or urine, on any study day were included, in adherence with the CDC guidance for ZIKV infection (https://www.cdc.gov/zika/laboratories/lab-guidance.html). The clinical assessments comprised past medical history, symptoms assessment, complete physical including neurological exam, disability assessment, assessment of complications including hospitalization, and assessment of presence of Guillain-Barre syndrome, with dedicated testing, if present[32]. Serum and urine were tested on days 0, 3, and 7 by PCR for ZIKV, DENV, and Pan-flavivirus[20]. The age, gender, days since onset of illness, prior DENV exposure, PCR testing for Zika, dengue and Pan-flavivirus (on both serum and urine samples) and ELISA testing (IgG and IgM antibodies) for ZIKV and DENV of these subjects (all Hispanic ethnicity) are detailed in Supplementary Table 1. The clinical symptoms experienced by each patient are documented in Supplementary Table 2. Convalescent dengue samples were collected in 2012 from a separate cohort study conducted in Cuernavaca, Mexico (prior to emergence of ZIKV in Mexico). Convalescent serum samples from ZIKV-infected individuals were obtained from BEI Resources, NIAID, NIH. Researchers performing antibody assays were blinded to the identity of samples.

**Clinical laboratory assays**. Serologic assays were performed for ZIKA IgG and IgM antibodies by ELISA (Euroimmun), DENV IgG, and IgM antibodies by ELISA (Panbio) using commercial diagnostic kits. PCR assays for Zika[20] (recommended by WHO), dengue[50], and Pan-flavivirus[51] were performed as described in the respective publications.

**ZIKV whole-genome fragment phage display library (ZIKV-GFPDL) construction**. Plasmid (ZIKV-ICD) containing complementary DNAs complementary to the whole genome of Paraiba_01/2015 strain of ZIKV was a gift from Alexander G. Pletnev, NIAID, NIH. This strain is closely related to circulating ZIKV strains in Mexico in the current study. A gIII display-based phage vector, fSK-9-3, where the desired polypeptide can be displayed on the surface of the phage as a gIII-fusion protein, was used for construction of the ZIKV whole-genome fragment phage display library (ZIKV-GFPDL). Purified PCR amplified DNA of whole genome of ZIKV (Supplementary Fig. 1) amplified using ZIKV-ICD was digested with *DNase*I to obtain gene fragments ranging in size from 200 to 1000 bp, and used for GFPDL construction by cloning in the fSK-9-3 phage vector for expression as gIIIp fusion protein, followed by electroporation into *E. coli* TG1 cells. Tet-resistant transformants were harvested and expanded in liquid culture (2X-YT medium) at 37 °C. The cell-free phage supernatant was isolated by centrifugation and phage titer was determined as Tet$^r$ transduction units[23]. PCR-based sequencing of individual clones was performed to ascertain the random distribution of both size and sequence of peptide displayed on the phage surface (Supplementary Fig. 2). Commercially available conformation-dependent MAbs, ZV54 (Millipore, Catalog # MABF2046), ZV67, Z23, and ZKA64 (Absolute Antibody; catalog # Ab00812-23.0, Ab00941-23.0, and Ab00779-23.0, respectively) were used for GFPDL characterization.

**Adsorption of polyclonal human sera on ZIKV-GFPDL phages and residual reactivity to ZIKV-E**. To demonstrate the capacity of the ZIKV-GFPDL to remove anti-ZIKV antibodies, 500 μl of 10-fold diluted serum antibodies from five post-infection human sera were adsorbed by incubation with ZIKV-GFPDL phage-coated Petri dishes. To ascertain the residual antibodies binding capacity, an SPR was performed with GLC chips coated with 500 RU of recombinant ZIKV-E. Human serum (with or without ZIKV-GFPDL adsorption) in Bovine serum albumin (BSA)-PBST buffer (phosphate-buffered saline (PBS) pH 7.4 buffer with Tween-20 and BSA) were injected at a flow rate of 50 μL/min (240-s contact duration) to determine the total anti-ZIKV-E binding antibodies before and after adsorption on ZIKV-E GFPDL.

**Adsorption of polyclonal post-infection human sample on EBOV GFPDL phages and residual reactivity to ZIKV particles**. Prior to panning of GFPDL, 500 μl of 10-fold diluted serum antibodies from pooled post-infection serum samples collected at day 7 visit was adsorbed by incubation with ZIKV-GFPDL phage-coated Petri dishes. To ascertain the residual antibodies specificity, an ELISA was performed with wells coated with heat inactivated $10^4$ TCID$_{50}$/100 μl of ZIKV, PRVABC59 virus particles. After blocking with PBST containing 2% milk, serial dilutions of human serum sample (with or without adsorption) in blocking solution were added to each well, incubated for 1 h at room temperature (RT), followed by addition of 5000-fold diluted horseradish peroxidase (HRP) -conjugated goat anti-human IgA+IgG+IgM-specific antibody and developed by 100 μl of O-phenylenediamine (OPD) substrate solution. Absorbance was measured at 490 nm. The absorbance values for test samples were subtracted from "virus only" negative control absorbance values.

**Affinity selection of ZIKV-GFPDL with serum and urine samples**. For GFPDL analysis, serum (and corresponding urine samples) were pooled from five acutely ZIKV-infected patients (highlighted in yellow in Supplementary Table 1). Prior to panning, serum or urine components that could nonspecifically interact with phage proteins were removed by incubation in UV-killed M13K07 phage-coated Petri dishes. Equal volumes of 10-fold diluted pooled polyclonal human sera (day 0 or day 7) or urine (day 7) were used for each round of GFPDL panning. GFPDL affinity selection was carried out in solution with protein A/G beads (for IgG) or IgM-specific capture beads to define the fine epitope specificity of these polyclonal IgG and IgM isotype antibodies. The unbound phages were removed by PBST (PBS containing 0.1% Tween-20) wash followed by washes with PBS. Bound phages were eluted by addition of 0.1 N Gly-HCl pH 2.2, and neutralized by adding 8 μl of 2 M Tris solution per 100 μl eluate[23,24]. After panning, antibody-bound phage clones were amplified, the inserts were sequenced, and the sequences were aligned to the ZIKV genome. Subsequently, additional IgM and IgG antibody epitope repertoire analysis was performed with serum sample of an acutely ZIKV-infected individual (Patient # 42-001-F) at day 7 visit (day 7 since onset of symptoms) to define the fine epitope specificity in an individual. This individual was part of the five pooled samples used for GFPDL analysis in Figs. 1 and 2. GFPDL affinity selection experiments were performed in duplicate (two independent experiments by research fellow, who was blinded to sample identity) and showed similar numbers of phage clones and epitope repertoires.

**Protein alignment of ZIKV and other flaviviruses**. An alignment of various ZIKV strains [Paraiba/2015 (GenBank#ANT96596.1 MR-766/Uganda/1947 (GenBank#ANK57895.1), Nigeria/IbH-30656_SM21V1-V3/1968 (GenBank#AMR68906.1), ArD157995/Senegal/2001 (GenBank#AHL43503.1), Micronesia/2007 (GenBank#ACD75819.1) and Brazil/2015 (GenBank#AMD16557.1)] and flaviviruses [dengue 1 (NCBI Reference#NP_059433.1), dengue 2

(NCBI Reference#NP_056776.2), dengue 3 (NCBI Reference#YP_001621843.1), dengue 4 (NCBI Reference#NP_073286.1), West Nile virus (UniProt Accession#P06935), and Yellow Fever virus/17D, (UniProt Accession#P03314)] was performed using MUSCLE program prior to generating heat maps showing conservation and similarity plots (SimPlot). Using Simplot, a query sequence (ZIKV_Paraiba strain) was used to generate a plot that will show the percent similarity of the reference sequences to other flaviviruses or to other ZIKV strains. A sliding window of size 200 bp or 20 bp was used, with the alignment in steps of 1 bp to generate the SimPlot showing different flaviviruses and all ZIKV strains, respectively.

**Surface representation of antigenic sites of various ZIKV proteins**. The crystal structures of various proteins that are a part of ZIKV genome E [PDB# 5U4W (immature E), 5JHM (mature E)], NS1 (PDB# 5K6K), NS2B (PDB# 5GXJ), NS3 (PDB# 5JRZ), and NS5(PDB# 5TFR) were used to depict surface representation of sequence conservation and antigenic sites pertaining to each of these ZIKV proteins.

**ELISA**. Biotinylated peptides (200 ng/well) were captured onto wells coated with 200 ng of streptavidin. Following three washes with phosphate buffered saline containing Tween-20 (PBST) (20 mM PBS, 0.1% Tween-20) plates were blocked with PBST containing 5% BSA (BSA-PBST). For testing, all specimens were five-fold serially diluted (starting at 1:100) in BSA-PBST and added to peptide-coated wells for 1 h at RT in duplicate. After three washes with PBST, the wells were reacted with HRP-conjugated goat anti-human IgG-A-M antibody (diluted 1:2000) (Jackson ImmunoResearch, West Grove, PA) at RT for 1 h, followed by addition of OPD substrate.

The cut-off values used are twice the average absorbance of negative control sera (at similar dilutions) for each peptide.

**Real-time antibody binding kinetics of post-ZIKV-infected human sera or urine samples to recombinant ZIKV-E, Domain III and NS1 proteins by SPR**. Steady-state equilibrium binding of longitudinal samples post-ZIKV-infected human polyclonal sera or urine from every individual in the clinical study was monitored at 25 °C using a ProteOn surface plasmon resonance (Bio-Rad). The purified recombinant ZIKV proteins (ZIKV-E and ZIKV-E-Domain III from Sino Biologicals and ZIKV-NS1 from Meridian Life Sciences) were coupled to a GLC sensor chip via amine coupling with either 100 or 500 RUs in the test flow channels. The protein density on the chip was optimized such as to measure only monovalent interactions independent of the antibody isotype. Importantly, the kinetics of disassociation was identical irrespective of the serum dilution and MAX RU (Supplementary Fig. 16). Samples of 300 μl freshly prepared sera at 10-fold and 40-fold dilution in BSA-PBST buffer (PBS pH 7.4 buffer with Tween-20 and BSA) were injected at a flow rate of 50 μl/min (120-s contact duration) for association, and disassociation was performed over a 600-s interval. Responses from the protein surface were corrected for the response from a mock surface and for responses from a buffer-only injection. SPR was performed with serially diluted serum (10-fold, 40-fold) or of urine of each individual participant in this study such that the SPR signal of the samples between 10 and 100 RU was used for further quantitative analysis. The Max RUs for each sera and urine sample in the manuscript figures was calculated by multiplying the observed RU signal with the dilution factor to provide the data for undiluted serum/urine sample. Total antibody binding and antibody isotype analysis were calculated with BioRad ProteOn manager software (version 3.1.0). All SPR experiments were performed twice, and the researchers performing the assay were blinded to sample identity. In these optimized SPR conditions, the variation for each sample in duplicate SPR runs was <5%.

Antibody off-rate constants, which describe the stability of the complex, i.e., the fraction of complexes that decays per second, were determined directly from the post-ZIKV-infected human polyclonal serum or urine sample interaction with ZIKV proteins using SPR (as described above) and calculated using the BioRad ProteOn manager software for the heterogeneous sample model[25].

**Statistical analyses**. The statistical significances between each time point (visit days) were determined using two-tailed paired *t*-test in GraphPad software. One-way analysis of variance (ANOVA) was performed with a Bonferroni post-hoc analysis for ELISA data. Correlations were calculated with a Spearman's test. *p*-Values <0.05 were considered significant with a 95% confidence interval.

**Ethics statement**. The study at CBER, FDA was conducted with de-identified samples under FDA Research Involving Human Subjects (RIHSC) exemption #17–050B; and all assays performed fell within the permissible usages in the original consent.

**Reporting summary**. Further information on research design is available in the Nature Research Reporting Summary linked to this article.

## Data availability

The data sets generated during and/or analyzed during the study are available from the corresponding author upon reasonable request. The source data underlying Figs. 1, 2, 5, 6, and 7 are provided in Supplementary Tables 1–4 and as a Source Data file.

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

## Acknowledgements

We thank Keith Peden and Marian Major for their insightful review of the manuscript. We thank A. Pletnev for generous gift of ZIKV-ICD plasmid. The La Red Zik01 study sites and staff instrumental in conducting the study and collecting the samples include: Hospital Regional de Alta Especialidad Ciudad Salud (HRAE) Tapachula: José M. Pérez, Gabriel Nájera, Karina Trujillo, Alfredo Vera, Alexander López; Instituto de Seguridad y Servicios Sociales de los Trabajadores del Estado (ISSSTE) Tapachula: Sandra Caballero, Zoila Priego, Francisco Camas; Instituto Mexicano del Seguro Social (IMSS) Tapachula: Héctor Rincón Karla Navarro, Antonia Sumuano, Cielo Mayorga; Instituto Nacional de Nutricion Salvador Zubiran (INNSZ): Alfredo Ponce de Leon, Miriam Bobadilla, Ernesto Maravilla, Luz Elena Cervantes, Fernando Arteaga, Fernando Ledesma, Pilar Ramos, Gustavo Rosales, and Alberto Garcia. La Red is collaboration between the Mexico Ministry of Health and the U.S. National Institute of Allergy and Infectious Diseases. This study has been funded in part by CONACYT (Fondo Sectorial SSA/ IMSS/ISSSTE, Projects No. 71260 and No. 127088). This study has been funded in part by the Division of Clinical Research, National Institute of Allergy and Infectious Diseases, National Institutes of Health. This project has been funded in part with federal funds from the

National Cancer Institute, National Institutes of Health, under Contract No. HHSN261200800001E and HHSN261201500003I. The antibody characterization work described in this manuscript was supported by FDA Medical Countermeasures Initiative (MCMi) funds to S.K. The latter funders had no role in study design, data collection and analysis, decision to publish, or preparation of the manuscript. The content of this publication does not necessarily reflect the views or policies of the Department of Health and Human Services, nor does mention of trade names, commercial products, or organizations imply endorsement by the U.S. Government.

## Author contributions

Designed research: S.K. and J.H.B. Performed research: S.R., M.H., and S.K. Conducted clinical study, contributed clinical samples, and provided clinical data: P.F.B.-Z., J.R.-C., G.N.-C., S.C.-S., K.R.N.-F., G.R.-P., and J.H.B. Contributed to writing: S.K., H.G., P.F.B.-Z., and J.H.B.

## Additional information

**Competing interests:** The authors declare no competing interests.

