## [Peer Review File · Nature Communications]

Reviewers' comments:

Reviewer #1 (Remarks to the Author):

In this study, the authors used the "GFPDL" strategy previously developed by their group to analyze the differential evolution of IgG and IgM repertoires in serum and urine from acutely ZIKV infected individuals. They found that epitope repertoire of IgM and IgG antibodies was variable post ZIKV infection. Furthermore, differential evolution of affinity maturation was observed for antibodies against structural and non-structural proteins in different body fluids following ZIKV exposure. These findings are quite interesting and could extend our understanding on antigenic epitopes of antibodies following ZIKV infection; the identified new antigenic sites especially NS2B peptides have the potential to be further evaluated as serodiagnostic targets and developed as effective countermeasures against ZIKV disease. Overall it is a straightforward and useful investigation with well-designed and carefully conducted experiments. The manuscript should be carefully examined for typos, organization of some sentences, and grammar.

Below are some suggestions for consideration,

1. The specificity of ZIKV-GFPDL should be tested against serum and urine from healthy human.
2. Equal volumes of pooled polyclonal human sera (day 0 or day7) or urine (day7) were used for each round of GFPDL panning. Sera contain many proteins in addition to antibodies, and these proteins may have non-specific bind to phages. Perhaps using equal amount of purified polyclonal IgM or IgG from serum or urine for panning could be more appropriate? Can explain.
3. The authors also found unlinked evolution of human antibody repertoires to different proteins during some other viral infections, e.g., RSV, influenza. Please discuss the differences in the findings with ZIKV infections and potential underlying mechanisms.
4. Line 28: the number of the keywords is suggested to be 3-8.
5. Line 40-43: the logicity in the first two sentences should be strengthened.
6. Line 64: use "monoclonal antibodies" instead of MAbs for its first appearance in the manuscript.
7. Line 67: replace "Flaviviruses" with "flaviviruses" throughout the manuscript.
8. Line 70: "Whole-Genome-Fragment-Phage-Display Libraries (GFPDL) has... , the symbol " is missing.
9. Line 109: sequence "of" the entire
10. Line 112: change "panel of" to "a panel of"
11. Line 121: change "Zika-GFPDL" to "ZIKV-GFPDL"
12. Line 166-169: according to Fig1B, the distribution of inserts recognized by urine IgM antibodies was similar to the binding pattern of the serum IgM antibodies binding to the antigenic sites in NS3, but not similar in NS5. The authors should correct this point.
13. Line 186: "Mature" to "mature"
14. Line249-255: these results are more relevant to the title "Conservation of antigenic sites with ZIKV strains and other Flaviviruses" instead of the "Serodiagnostic potential of GFPDL selected antigenic site peptides". I suggest that authors should carefully consider how to organize this part of the results.
15. Line 280: "and 42-2) ," -- delete the space

16. Line432: as the authors mentioned "the numbers of study participants are limited to draw strong conclusions", why didn't adopt more participants in investigating the correlation between the day 7 antibody binding affinity against the E-ectodomain and the number of clinical symptoms on day 28 post-ZIKV infection?

17. The image resolution is too low in most figures and supplementary figures containing ZIKV ICD translated sequence. The requirements of photographic and bitmapped images can be found in the web <https://www.nature.com/ncomms/submit/how-to-submit>. Figure 2, a better presentation of the panel A is needed, as the characters are too small. Figure 6: The significance between groups should be marked with an asterisk.

Reviewer #2 (Remarks to the Author):

The manuscript submitted by Ravishandran et al presents a detailed analysis of serum and urine analyses of antibody responses after ZIKV infection, using a phage display approach. While the manuscript presents a very interesting set of measurements, several technical issues with this approach need to be more thoroughly addressed throughout.

The sequence alignments presented and how the authors findings fit with this is excellent!

The potential application of the authors findings for diagnostic purposes are very useful, but could be expanded upon experimentally.

The authors acknowledge that one weakness with this peptide based approach is that conformational epitopes may not be completely represented in their library. The authors present a set of experiments addressing this issue, but concerns remain despite this. These concerns are especially significant given that many neutralizing antibodies have been described that recognize quaternary epitopes. The authors show a small number of previously described antibodies that still can recognize phages in the libraries. In addition, the authors also show that depletion of the serum/urine samples with phage panning leads to an almost complete removal of binding to recombinant E protein. These experiments should be complemented with testing of binding and neutralization activity of the depleted serum samples to confirm that the assay test for a majority of relevant epitope specificities. This is especially important as the authors state that several donors fail to bind to E or NS1. Is this really so., or simply that these donors respond primarily to quaternary epitopes?

All clinical binding data and virus neutralizing titers should be included in the manuscript.

It is not clear what the relevance of the urine analysis is. Existing data on antibodies in urine, and their relevance for understanding protective immunity should be discussed.

For both urine and serum analyses the study would benefit of including flavivirus naive serum samples. This would serve as a control for background binding of serum antibodies to the ZIKV phage libraries. This would be important to understand especially the IgM responses, as there avidity would also play a role, possibly allowing for detection of very low affinity interactions.

It is confusing that there is not correlation between the reactivity of the urine antibodies and serum antibodies. What does the urine antibodies really mean and why is there no baseline data provided for this set?

Overall the naming convention is confusing, where Day 0 represents the first sample obtained, ranging from the day of symptom onset and several days later.

The authors mention that the donors analyzed live in a dengue endemic area, yet there was only one volunteer patient that had a documented dengue exposure. It is unclear however if this means that the authors think that all participants were dengue naive? The discussion would benefit from presenting data on dengue seroprevalence in this region, and the likelihood of previous,

unreported exposure.

Finally a major shortcoming is that pooled sera and urine is used for these analyses. While it is appreciated how labor intensive this approach is, the fact that pooled sera and urine are used, reduce the potential for these data to understand the complexity of individual responses. At least a figure panel should be included showing ZIKV binding (IgG and IgM) and virus neutralization of each individual sera/urine (best for all, but at least for the 5 donors included in the pooled samples) to better understand the contribution of each sample to the overall signal observed. Is the data representative primarily of one potentially responding donor, or does all five samples contribute equally?

Figure S2 shows that the peptide library is distributed over the entire cDNA clone, but the lack of depth for this analysis makes it unclear if there are "holes" in the library. For example, this figure (and figure 2) suggest that there are no peptides covering the fusion loop in the library. It seems unlikely from existing literature that none of the donors would recognize this epitope.

Further discussion of the symptoms seen at day 28, possibly correlating with the affinity maturation analysis at day 7. It is also not clear why day 7 is used to determine affinity maturation. This mechanism is generally thought to happen through germinal center selection and I would be surprised to see much of this in such a short timeframe.

The off-rate analysis is only presented in bulk. Would be more informative to show this as a paired statistical analysis. Also, the authors mention in the methods that two concentrations are measured, suggesting that the overall concentration of antibody would have an impact on the off rate measurement. Additional analyses illustrating that the minor effects observed are not simply a correlate for magnitude of response should be included.

Minor points:

What does the "total %" in supplementary table 3 signify?

Response to Reviewers :

Reviewer #1:

In this study, the authors used the “GFPDL” strategy previously developed by their group to analyze the differential evolution of IgG and IgM repertoires in serum and urine from acutely ZIKV infected individuals. They found that epitope repertoire of IgM and IgG antibodies was variable post ZIKV infection. Furthermore, differential evolution of affinity maturation was observed for antibodies against structural and non-structural proteins in different body fluids following ZIKV exposure. These findings are quite interesting and could extend our understanding on antigenic epitopes of antibodies following ZIKV infection; the identified new antigenic sites especially NS2B peptides have the potential to be further evaluated as serodiagnostic targets and developed as effective countermeasures against ZIKV disease. Overall it is a straightforward and useful investigation with well-designed and carefully conducted experiments.

Response: We highly appreciate the positive comments and acknowledging the importance of this study.

Below are some suggestions for consideration,

1. The specificity of ZIKV-GFPDL should be tested against serum and urine from healthy human.

Response:

See modified Figure 1 and text (Results; lines 155-159):

“As a negative control we used a serum from flavivirus naive individual. This serum bound very few phages of the ZIKV-GFPDL (412 and 103 phages bound by IgM and IgG antibodies, respectively) (Fig. 1A). Sequencing of these bound phage clones showed random distribution across the entire ZIKV genome both for IgM and IgG antibody profile (Fig. 1 B, C marked ‘Naïve serum’).”

We highlighted the 5 patients used for the GFPDL analysis in Suppl. Table 1.

2. Equal volumes of pooled polyclonal human sera (day 0 or day7) or urine (day7) were used for each round of GFPDL panning. Sera contain many proteins in addition to antibodies, and these proteins may have non-specific bind to phages. Perhaps using equal amount of purified polyclonal IgM or IgG from serum or urine for panning could be more appropriate?

Response:

The GFPDL panning has been carefully developed and optimized over many years to address such parameters. To remove any non-specific interactions of serum proteins with the phages, all samples are first adsorbed twice with control wild type M13K07 phages

(no inserts) to remove non-specific binding activities as specified in Methods (lines 574-576):

“Prior to panning, serum or urine components that could nonspecifically interact with phage proteins were removed by incubation in UV-killed M13K07 phage-coated petri dishes.”

It is not practical to isolate IgG and IgM from the low volume clinical samples. Furthermore, we believe the purification of IgG and IgM is not required as it is already achieved in our GFPDL panning protocol since it is performed in presence of protein A (IgG) and anti-IgM beads that serve similar function as antibody purification during affinity selection of phages. All the ZIKV-GFPDL antibody-bound phages are selected with beads coated with Protein A or anti-IgM antibodies. Therefore, all the GFPDL binding data presented are only for IgM and IgG antibodies in the serum/urine samples.

See Methods (lines 577-580):

“GFPDL affinity selection was carried out in solution with protein A/G beads (for IgG) or IgM-specific capture beads to define the fine epitope specificity of these polyclonal IgG and IgM isotype antibodies as previously described.”

3. The authors also found unlinked evolution of human antibody repertoires to different proteins during some other viral infections, e.g., RSV, influenza. Please discuss the differences in the findings with ZIKV infections and potential underlying mechanisms.

Response: (added to Discussion) (lines 403-414):

“We previously reported unlinked evolution of antibody binding to subdomains in the influenza hemagglutinin (HA1 vs. HA2) and RSV membrane proteins (F vs. G) overtime. The unlinked antibody evolution against these viral proteins in influenza and RSV is most likely due to multiple mechanisms including differential viral protein expression following virus infection, protein secretion/release from infected cells, antigen presentation, immune-dominance, pre-existing immunity and immune selection over time. In contrast, for flaviviruses, a single polyprotein is being cleaved to generate the structural and non-structural proteins at equimolar amounts. In this study, we measured immune responses against more conserved and less conserved proteins shortly after primary acute ZIKV infection. During that time frame, it is unlikely that prior immunity played a key role in the observed differential antibody responses.”

4. Line 28: the number of the keywords is suggested to be 3-8.

Response: (line 27-8)

Modified key words: Zika, Infection, Vaccine, Immune Response, Antibody, Epitope mapping, Affinity, Diagnostics.

5. Line 40-43: the logicity in the first two sentences should be strengthened.

Response (modified Abstract): (lines 39-43)

“The outbreak of Zika virus (ZIKV) infections in the Americas linked to microcephaly in newborns and other neurological complications has led to extensive efforts to develop vaccines and ZIKV-specific early diagnostic tests. These efforts can greatly benefit from information on the repertoire of antibody response generated following acute ZIKV infection and its evolution over time.”

6. Line 64: use “monoclonal antibodies” instead of MAbs for its first appearance in the manuscript.

Response: Suggested change inserted

7. Line 67: replace "Flaviviruses" with "flaviviruses" throughout the manuscript.

Response: Suggested change in spelling inserted throughout the manuscript

8. Line 70: “Whole-Genome-Fragment-Phage-Display Libraries (GFPDL) has... , the symbol ” is missing.

Response: (line 70)

Changed to: “Whole-Genome-Fragment-Phage-Display-Libraries” (GFPDL)”

9. Line 109: sequence "of" the entire

Response: Correction inserted

10. Line 112: change "panel of" to "a panel of"

Response: Correction inserted

11. Line 121: change "Zika-GFPDL" to "ZIKV-GFPDL"

Response: Correction inserted

12. Line 166-169: according to Fig1B, the distribution of inserts recognized by urine IgM antibodies was similar to the binding pattern of the serum IgM antibodies binding to the antigenic sites in NS3, but not similar in NS5. The authors should correct this point.

Response (modified text in Results): (lines 179-182)

“The distribution of inserts recognized by urine IgM antibodies (from day 7 visit) was similar to the binding pattern of the serum IgM antibodies with predominant binding to the antigenic sites in NS3, but not similar in NS5 (Fig. 1B, Fig. 2B and Suppl. Table 3).”

13. Line 186: "Mature" to "mature"

Response: Correction inserted

14. Line249-255: these results are more relevant to the title “Conservation of antigenic sites with ZIKV strains and other Flaviviruses” instead of the “Serodiagnostic potential of GFPDL selected antigenic site peptides”. I suggest that authors should carefully consider how to organize this part of the results.

Response: The paragraph was moved to conservation section (lines 244-248)

“Several sequences in the non-structural proteins (NS1, NS2B, NS3, NS4B, and NS5) recognized at high frequencies by the IgM and IgG antibodies in the serum or urine post-ZIKV infection and were highly conserved among different ZIKV strains could potentially be used for serodiagnosis of ZIKV infection (Suppl. Fig. 14 and Suppl. Table 4).”

15. Line 280: *"and 42-2) ," -- delete the space*

Response: Correction inserted

16. Line 432: *as the authors mentioned “the numbers of study participants are limited to draw strong conclusions”, why didn’t adopt more participants in investigating the correlation between the day 7 antibody binding affinity against the E-ectodomain and the number of clinical symptoms on day 28 post-ZIKV infection?*

Response: We fully agree that the correlation between antibody affinity maturation and resolution of clinical symptoms should be followed up for confirmation using larger cohort. However, this study was put together and initiated at the late stages of the ZIKV outbreak in Mexico, when ZIKV infection were declining and no additional acute ZIKV infection longitudinal samples were available.

17. The image resolution is too low in most figures and supplementary figures containing ZIKV ICD translated sequence. The requirements of photographic and bitmapped images can be found in the web <https://www.nature.com/ncomms/submit/how-to-submit>. Figure 2, a better presentation of the panel A is needed, as the characters are too small. Figure 6: The significance between groups should be marked with an asterisk.

Response: Figure image resolution was enhanced (Figures 1 and 2). Significance in Figure 6 is now shown with an asterisk.

Reviewer #2:

The manuscript submitted by Ravishandran et al presents a detailed analysis of serum and urine analyses of antibody responses after ZIKV infection, using a phage display approach. While the manuscript presents a very interesting set of measurements, several technical issues with this approach need to be more thoroughly addressed throughout.

The sequence alignments presented and how the authors findings fit with this is excellent!

The potential application of the authors findings for diagnostic purposes are very useful, but could be expanded upon experimentally.

Response: We highly appreciate the positive comments and acknowledging the importance of this study.

1. The authors acknowledge that one weakness with this peptide based approach is that conformational epitopes may not be completely represented in their library. The authors present a set of experiments addressing this issue, but concerns remain despite this. These concerns are especially significant given that many neutralizing antibodies have been described that recognize quaternary epitopes. The authors show a small number of previously described antibodies that still can recognize phages in the libraries. In addition, the authors also show that depletion of the serum/urine samples with phage panning leads to an almost complete removal of binding to recombinant E protein. These experiments should be complemented with testing of binding and neutralization activity of the depleted serum samples to confirm that the assay test for a majority of relevant epitope specificities. This is especially important as the authors state that several donors fail to bind to E or NS1. Is this really so., or simply that these donors respond primarily to quaternary epitopes?

Response: As suggested by the reviewer, we performed additional ELISA experiment with whole ZIKV particles with GFPDL adsorbed serum sample that confirmed >90% removal of ZIKV specific binding antibodies in whole virion ELISA (Supplementary Figures 8). We also added epitope mapping data for new ZIKV-E specific conformation dependent neutralizing MAb (Supplementary Figures 3).

Please see the modified Results sections (Supplementary Figures 3-8): Line 112-141:

“To ascertain that the GFPDL represents both linear and conformational epitopes, we performed three independent experiments. Firstly, the ZIKV-GFPDL was used to map epitopes of a panel of linear and conformation dependent MAbs. GFPDL mapping data is shown for four representative previously described ZIKV-protective MAbs. ZV54 is ZIKV-specific neutralizing MAb against African, Asian, and American strains to varying degrees. Structurally it binds the lateral ridge in DIII of the envelope protein similar to MAb ZV67 (Suppl. Fig. 3). MAb ZV67 is a cross-reactive neutralizing and protective mouse MAb recognizing conformational epitope in the lateral ridge of E domain III32 (Suppl. Fig. 4). MAb Z23 is DENV-negative, ZIKV-specific neutralizing and protective human MAb that recognizes a conformation-dependent tertiary epitope in E domain III; mainly binds to one envelope protein monomer and can interact with two envelope protein dimers on the virion surface (Suppl. Fig 5), MAb ZKA64 (neutralizing and protective human MAb that recognizes a conformational epitope in E domain III) (Suppl. Fig. 6). These MAbs were derived from immune B cells following mouse immunization, or from ZIKV infected survivors.

For all four MAbs, strong binding to the ZIKV-GFPDL was observed (Suppl. Figs. 3-6). Importantly, the consensus epitope sequences obtained through GFPDL analysis were very similar to the footprints previously identified for these MAbs. The initial binding was confirmed by phage-ELISA using three phages expressing overlapping sequences for each of the MAbs (Suppl. Figs. 3-6, panel B). These results provided proof of concept that ZIKV-GFPDL approach can identify conformational epitopes recognized by previously described protective ZIKV-E specific MAbs.

Secondly, we determined the capacity of the phage display library to adsorb ZIKV-E specific antibodies in the post-ZIKV infected polyclonal human sera and urine. After two

rounds of adsorption with the ZIKV-GFPDL, pooled day 7 serum or urine samples demonstrated >90% removal of total anti-ZIKV-E binding antibodies as measured by Surface Plasmon Resonance (Suppl. Fig. 7). Thirdly, reactivity of the GFPDL-adsorbed sera was evaluated against the ZIKV particles in ELISA, that showed >90% of serum antibodies were adsorbed by the ZIKV-GFPDL (Suppl. Fig. 8). Together, these preliminary studies provided strong rationale of using this GFPDL for epitope mapping of post-ZIKV infection polyclonal sera/urine antibody repertoire.”

2. All clinical binding data and virus neutralizing titers should be included in the manuscript.

Response: 1.

Complete information on Gender, Age, sample collection days, days since onset of symptoms and clinical symptoms for all ZIKV infected patients in the current study are shown in Supplementary Table 2.

Unfortunately, we did not have a virus neutralization assay setup and this testing has not been performed on these samples. Complete patients’ information for all time points and samples (PCR testing of both serum and urine for Zika, Dengue and Pan-flavivirus virus, commercial ELISA binding of IgG and IgM antibodies to both Zika and Dengue virus, number of clinical symptoms) and potential confounders for all samples used in the current study are shown in Supplementary Table 1. The 5 individuals that were used for GFPDL analysis are also highlighted in yellow to provide complete testing information about these samples.

See modified Results section (lines 144-163)

“For GFPDL analysis, serum (and corresponding urine samples) were pooled from 5 acutely ZIKV infected patients (highlighted in yellow in Suppl. Table 1). Subjects 41-010-F, 41-017-F, 42-001-F, 42-003-F, and 42-018-F) gave PCR positive results for ZIKV RNA in serum and/or urine on visit 1. They were 1-3 days post onset of symptoms. In ELISA these patients had low anti-ZIKV reactivity of IgM (O.D. <0.1) and IgG (O.D. <0.4) antibodies that increased in titers on subsequent visits. In all cases, low to moderate reactivity was also observed in dengue virus ELISA, confirming the known cross-reactivity of antibodies between these closely related flaviviruses (Suppl. Table 1). While none of these 5 subjects reported prior Dengue virus infection, we cannot exclude the possibility of prior Dengue virus exposure based on the observed ELISA binding results and the high Dengue seroprevalence previously reported in the region.

As a negative control we used a serum from flavivirus naive individual. This serum bound very few phages of the ZIKV-GFPDL (412 and 103 phages bound by IgM and IgG antibodies, respectively) (Fig. 1A). Sequencing of these bound phage clones showed random distribution across the entire ZIKV genome both for IgM and IgG antibody profile (Fig. 1 B, C marked ‘Naïve serum’).”

(lines 294-304):

“However, 16% (3/19) of ZIKV-infected individuals did not show significant serum antibody binding (Max RU <20) to either the E or domain III proteins following ZIKV

exposure. Two of the three ZIKV-E non-responsive individuals were the same individuals that did not show reactivity to NS1 protein (samples 41-2 and 42-2), and the third individual had very low NS1 binding (sample 41-3). Interestingly, these three individuals gave positive reactivity in the commercial ZIKV-ELISA at the first visit but did not show increase in either IgG or IgM binding titers on subsequent visits (Suppl. Table 1). Only one of these individuals (42-2) acknowledged prior Dengue infection (Suppl. Table 1). However, we cannot exclude the possibility that these individuals developed some antibodies to quaternary epitopes in ZIKV-E, E-domain III, or NS1 that were not captured by our SPR binding assay.”

3. It is not clear what the relevance of the urine analysis is. Existing data on antibodies in urine, and their relevance for understanding protective immunity should be discussed. It is confusing that there is not correlation between the reactivity of the urine antibodies and serum antibodies. What does the urine antibodies really mean and why is there no baseline data provided for this set?

Response (Discussion): (lines 390-396)

“The urine day 7 IgM antibodies had similar, but not identical, repertoires to the serum IgM antibodies. The presence of IgM in urine can be explained by a urogenital ZIKV replication, in agreement with previous reports. Recent studies suggested that plasma and urine ZIKV PCR positivity are not linked, supporting local infections in different organs resulting in localized immune responses, in agreement with our findings. Recent studies suggested that plasma and urine ZIKV PCR positivity are not linked, supporting local infections in different organs resulting in localized immune responses, in agreement with our findings. These studies supported the use of urine samples, which is used for diagnosis of ZIKV infections.”

4. For both urine and serum analyses the study would benefit of including flavivirus naive serum samples. This would serve as a control for background binding of serum antibodies to the ZIKV phage libraries. This would be important to understand especially the IgM responses, as there avidity would also play a role, possibly allowing for detection of very low affinity interactions.

Response: We performed additional GFPDL panning with control serum from flavivirus naive individual and data has been added to Figure 1. Negative urine samples from flavivirus naive individual were not available.

Modified Fig. 1 and Results (lines 155-159):

“As a negative control we used a serum from flavivirus naive individual. This serum bound very few phages of the ZIKV-GFPDL (412 and 103 phages bound by IgM and IgG antibodies, respectively) (Fig. 1A). Sequencing of these bound phage clones showed random distribution across the entire ZIKV genome both for IgM and IgG antibody profile (Fig. 1 B, C marked ‘Naïve serum’).”

5. Overall the naming convention is confusing, where Day 0 represents the first sample obtained,

ranging from the day of symptom onset and several days later.

Response:

We clarified this point in the Results section and legend to Table 1 (lines 101-104):

“For simplicity, samples are referred by the visit day throughout the manuscript rather than days post onset of symptoms. For most individuals the first visit ranged between 0-5 days from the day of symptom onset.”

6. The authors mention that the donors analyzed live in a dengue endemic area, yet there was only one volunteer patient that had a documented dengue exposure. It is unclear however if this means that the authors think that all participants were dengue naive? The discussion would benefit from presenting data on dengue seroprevalence in this region, and the likelihood of previous, unreported exposure.

Response: We have added information and clarification at multiple places throughout the manuscript.

Results section: lines 144-154):

“For GFPDL analysis, serum (and corresponding urine samples) were pooled from 5 acutely ZIKV infected patients (highlighted in yellow in Suppl. Table 1). Subjects 41-010-F, 41-017-F, 42-001-F, 42-003-F, and 42-018-F) gave PCR positive results for ZIKV RNA in serum and/or urine on visit 1. They were 1-3 days post onset of symptoms. In ELISA these patients had low anti-ZIKV reactivity of IgM (O.D. <0.1) and IgG (O.D. <0.4) antibodies that increased in titers on subsequent visits. In all cases, low to moderate reactivity was also observed in dengue virus ELISA, confirming the known cross-reactivity of antibodies between these closely related flaviviruses (Suppl. Table 1). While none of these 5 subjects reported prior Dengue virus infection, we cannot exclude the possibility of prior Dengue virus exposure based on the observed ELISA binding results and the high Dengue seroprevalence previously reported in the region.”

Discussion section (lines 386-390):

“In Tapachula, Chiapas, Mexico, where the study was conducted, DENV is common, yet only one confirmed prior Dengue virus exposure was reported in the current study (Suppl. Table 1). This individual was not included in the samples for the GFPDL analysis. Therefore, we cannot exclude the possibility of unconfirmed prior exposure to other flaviviruses in the study subjects resulting in recall antibody responses shortly after acute Zika virus infection.”

7. Finally a major shortcoming is that pooled sera and urine is used for these analyses. While it is appreciated how labor intensive this approach is, the fact that pooled sera and urine are used, reduce the potential for these data to understand the complexity of individual responses. At least a figure panel should be included showing ZIKV binding (IgG and IgM) and virus neutralization of each individual sera/urine (best for all, but at least for the 5 donors included in the pooled samples) to better understand the contribution of each sample to the overall signal observed. Is

the data representative primarily of one potently responding donor, or does all five samples contribute equally?

Response: As suggested by the reviewer, we performed GFPDL analysis of IgG and IgM antibody repertoire recognized by an individual serum sample and this data has been added to new Suppl. Figure 9.

See modified Results section (lines 197-203) and new Suppl. Fig. 9:

“Subsequently, additional IgM and IgG antibody epitope repertoire analysis was performed with serum sample of an acutely ZIKV-infected individual (Patient # 42-001-F) at day 7 visit (day 7 since onset of symptoms) to define the fine epitope specificity in an individual. This individual was part of the 5 pooled samples used for GFPDL analysis in Figures 1 and 2. The epitopes recognized by IgM and IgG antibodies identified similar pattern to the pooled samples (Suppl. Fig. 9). Again, the IgG bound epitopes were more focused than the IgM repertoire, with immunodominance of NS2B and E antibody specificities.”

8. Figure S2 shows that the peptide library is distributed over the entire cDNA clone, but the lack of depth for this analysis makes it unclear if there are "holes" in the library. For example, this figure (and figure 2) suggest that there are no peptides covering the fusion loop in the library. It seems unlikely from existing literature that none of the donors would recognize this epitope.

Response:

See modified Figure 2 (improved image): the antibody reactivity against the fusion loop is covered by the antigenic site Z 4.2 (E-2) 365-411, suggesting that fusion loop is recognized by both IgG and IgM antibodies in the post-ZIKV infected serum in Fig. 2B and 2C. See suppl. Table 3 and 4 for additional details.

In addition, the serum samples were reacted with ZIKV virions before and after adsorption with the ZIKV-GFPDL confirming that GFPDL can capture >90 % antibody reactivity in post-ZIKV infection sera (new Suppl. Fig. 8):

(lines 137-145):

“Secondly, we determined the capacity of the phage display library to adsorb ZIKV-E specific antibodies in the post-ZIKV infected polyclonal human sera and urine. After two rounds of adsorption with the ZIKV-GFPDL, pooled day 7 serum or urine samples demonstrated >90% removal of total anti-ZIKV-E binding antibodies as measured by Surface Plasmon Resonance (Suppl. Fig. 7). Thirdly, reactivity of the GFPDL-adsorbed sera was evaluated against the ZIKV particles in ELISA, that showed >90% of serum antibodies were adsorbed by the ZIKV-GFPDL (Suppl. Fig. 8). Together, these preliminary studies provided strong rationale of using this GFPDL for epitope mapping of post-ZIKV infection polyclonal sera/urine antibody repertoire.”

9. Further discussion of the symptoms seen at day 28, possibly correlating with the affinity maturation analysis at day 7. It is also not clear why day 7 is used to determine affinity maturation. This mechanism is generally thought to happen through germinal center selection

and I would be surprised to see much of this in such a short timeframe.

Response We have expanded the Results section (lines 336-345):

“To determine the relevance of antibody affinity maturation with the clinical disease following ZIKV infection, Spearman correlations were determined for the polyclonal sera antibody off-rates to ZIKV-E protein and NS1 proteins on the day of peak antibody binding titer (day 7) vs. number of clinical symptoms on day 28 visit. The first visit date ranged from 0-5 days post onset of symptoms, and onset of symptoms can be 3-12 days after infection. While this is still an acute infection response with predominant IgM responses there was evidence of class switching and affinity maturation between visit day 0 and visit day 7. Therefore, it was reasonable to probe the correlation between early antibody affinity maturation and the reduction in clinical symptoms by day 28.”

10. The off-rate analysis is only presented in bulk. Would be more informative to show this as a paired statistical analysis. Also, the authors mention in the methods that two concentrations are measured, suggesting that the overall concentration of antibody would have an impact on the off rate measurement. Additional analyses illustrating that the minor effects observed are not simply a correlate for magnitude of response should be included.

Response: As suggested by the reviewer for figure 6 dataset, a two-tailed paired t-test statistical analysis was performed, and the significance values have been added to the Figure 6 and legend respectively and described in the statistical methods section.

See clarification in Results section (lines 309-321):

“To further evaluate antibody affinity maturation following ZIKV infection against structural ZIKV-E and non-structural NS1 protein, the antibody-antigen complex dissociation rates (off-rate constants) were determined as a surrogate for affinity maturation using SPR. Antibody dissociation kinetics of antigen-antibody complexes are independent of antibody concentration and provide a measure of overall average affinity of polyclonal antibody binding. To that end, serially diluted serum/urine at 10-, 40-, and 160-fold dilutions were injected at a flow rate of 50 μ L/min (120-sec contact time) for association, and dissociation was performed over a 600 second interval (at a flow rate of 50 μ L/min) on ZIKV proteins (Suppl. Fig. 16 A and B, respectively). Antibody off-rate constants, which describe the fraction of antigen-antibody complexes that decay per second, were determined directly from the plasma antibody interaction with GP in the dissociation phase only for the sensorgrams with maximum RU in range of 10-100 RU using BioRad Proteon SPR machine (Suppl. Fig. 16).”

Minor points:

11. What does the "total %" in supplementary table 3 signify?

Response: Legend to Suppl. Table 3: Frequency of antigenic sites for IgM and IgG antibodies in serum on day 0 and 7 and urine on day 7 post -ZIKV exposure.

The calculations of frequencies (%) were based on inserts that were isolated at least twice (or more).

Total percentage of clones are comprised of clones represented in these antigenic sites for the analyzed sample. The remaining clones are not represented by any unique antigenic site as clonal frequency is less than 2 for all analyzed samples

REVIEWERS' COMMENTS:

Reviewer #1 (Remarks to the Author):

All my concerns were nicely addressed in the revised manuscript.